# Unifying Stacking and Cascading for Efficient Ensemble Inference

**Ashwin Gerard Colaço** [1]   **Sharad Mehrotra** [1]   **Michael De Lucia** [2]   **Kevin Hamlen** [3]   **Murat Kantarcioglu** [4]
**Latifur Khan** [3]   **Ananthram Swami** [2]   **Bhavani Thuraisingham** [3]   **Unnat Jain** [1]

## Abstract

We introduce LazyStack, a method for efficient model ensemble inference. The core idea is intuitive: after each model executes, we check whether accumulated evidence is sufficient to exit confidently. Sometimes one model suffices; other times we aggregate predictions from several models via trained meta-learners before reaching confidence. Two insights make this work. First, most inputs follow only 3 to 8 execution trajectories. This reduces the training problem from exponential to linear: we learn aggregators only for these common paths, not all possible model combinations. Second, we formulate trajectory selection as an MDP and use value iteration to compute the optimal routing policy, which reveals counterintuitive model orderings. On intrusion detection, starting with a moderately expensive model outperforms starting with the cheapest, because its higher confidence enables earlier overall exit. Across vision, text, tabular, and LLM tasks, we achieve up to 38x speedup at 97%+ accuracy retention compared to a complete ensemble. The result: ensemble-quality predictions at cascade-level cost. Code and a project page are available at https://ashwincolaco.github.io/lazystack/.

## 1. Introduction

Ensemble methods face a tradeoff: combining predictions from multiple models improves accuracy but multiplies inference cost. Consider a simple example: classifying an image might take a cheap model 15ms but an expensive one 32ms. For intrusion detection, the gap can reach 65x between fast simple models (0.8ms) and slow complex ones

(52ms). Can we achieve ensemble-level accuracy without running all models on every input?

*Stacked generalization* (Wolpert, 1992) ("stacking") runs multiple diverse models on each input and trains a meta-model to combine their predictions (Figure 1, top, left). For CIFAR-100, a 6-model stacker achieves 78% accuracy, exceeding any single model, but requires executing all 6 models (105ms) for every prediction. *Cascades* take the opposite approach (Figure 1, top, middle): execute models sequentially from cheapest to most expensive, stopping when one is confident enough. This reduces latency but sacrifices accuracy, since exit decisions rely on single-model confidence. Recent methods (Kolawole et al., 2025; Chen et al., 2024a; Ong et al., 2025) share this limitation: when multiple models have executed, they either ignore the value of combining their predictions or combine them heuristically.

We introduce LazyStack, which applies stacking progressively throughout a cascade (Figure 1, top right). The key insight is training meta-learners called substackers for every prefix of the execution sequence. After each model executes, we aggregate all predictions seen so far and exit if the combined confidence is sufficient. This way, even partial execution benefits from ensemble-quality aggregation.

This presents two challenges that make the naive approach intractable. With $k$ models there are $2^k$ possible subsets requiring separate aggregators—64 for 6 models, over 1000 for 10. Training and storing substackers for all combinations is prohibitive. Second, traditional cascades use fixed orderings like cheapest-first, but optimal ordering is input-dependent and non-obvious: a slightly more expensive model might produce confident predictions that enable earlier stopping, reducing *total* cost despite higher upfront cost. No fixed heuristic captures this tradeoff.

LazyStack addresses both through two components, enabled by a key empirical finding: *trajectory concentration*. Despite $2^k$ possible execution paths, we find that 95%+ of samples across all 8 datasets follow just 3–8 trajectories. This collapse from exponential to near-constant was not predictable from prior work and is what makes progressive stacking practical. First, *trajectory discovery* formulates model selection as a Markov Decision Process where re-

[1]University of California, Irvine, USA [2]Army Research Laboratory, USA [3]University of Texas at Dallas, USA [4]Virginia Tech, USA. Correspondence to: Ashwin Gerard Colaço <acolaco@uci.edu>.

*Proceedings of the $43^{rd}$ International Conference on Machine Learning*, Seoul, South Korea. PMLR 306, 2026. Copyright 2026 by the author(s).

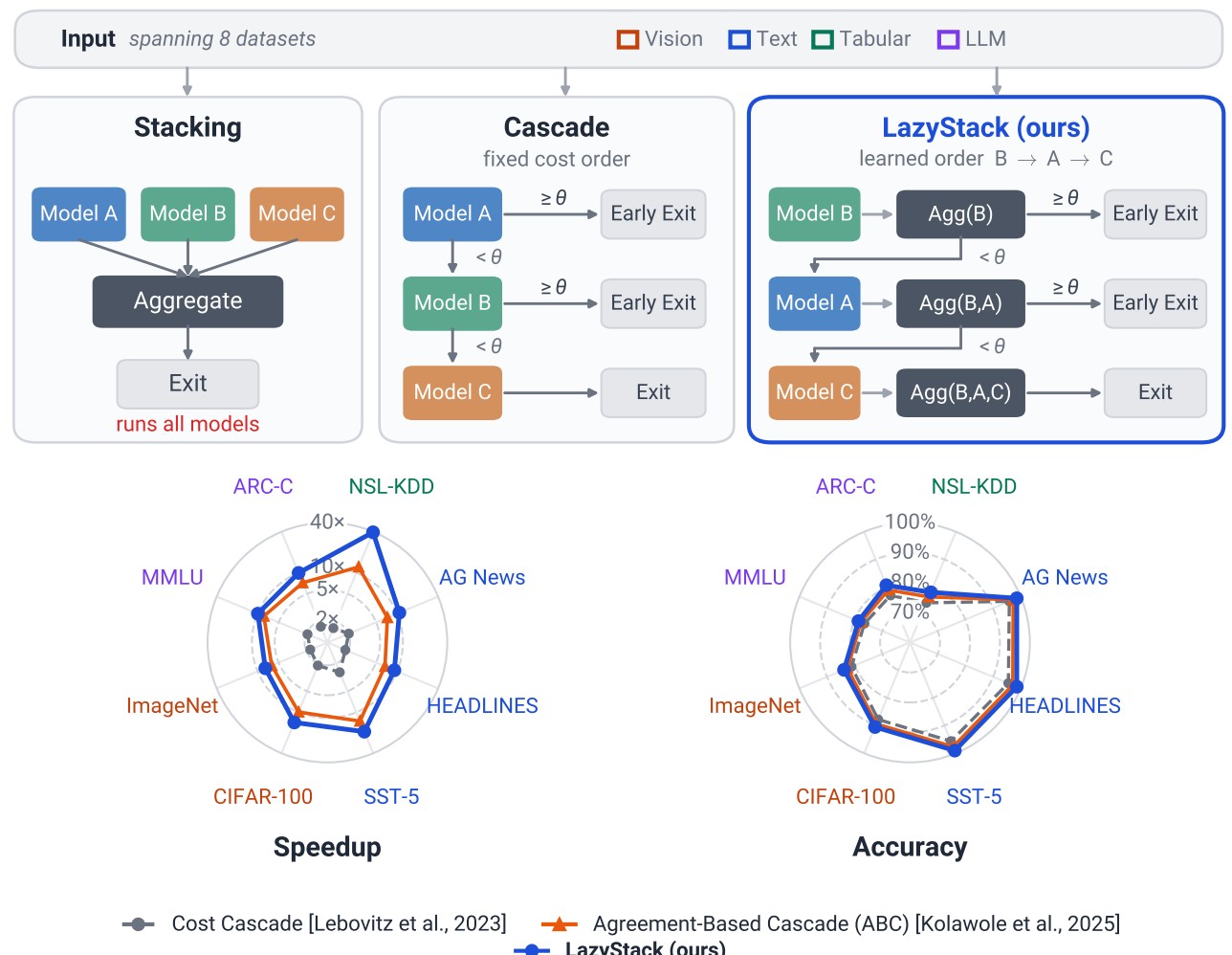

Figure 1. **Traditional Stacking and Cascades vs LazyStack's Progressive Stacking.** Top: Traditional stacking runs all models; cascades exit on single-model confidence without aggregation. LazyStack discovers model orderings via MDP and progressively combines predictions using trained substackers if available, enabling confident early exit from accumulated evidence. Bottom: Across 8 benchmarks, LazyStack matches baseline accuracy while achieving 2.5–38× speedup.

wards balance accuracy ($\beta$) against cost ($\alpha \cdot c_j$). Solving this MDP on validation data discovers 3 to 8 high-coverage trajectories representing effective execution patterns. Importantly, the MDP discovers sequences from data rather than heuristics. For example, on intrusion detection it learns that starting with LightGBM (2.1ms) outperforms the cheaper Logistic Regression (0.8ms), because LightGBM's higher confidence enables earlier exit that more than compensates for the extra upfront cost. This is counterintuitive. It contradicts the cheapest-first heuristic used by all prior cascade methods, yet emerges naturally from learned optimization.

Second, *prefix meta-learning* trains substackers for every prefix along discovered trajectories. For trajectory $[A, B, C]$, we train three substackers: one for A alone, one for A+B, one for A+B+C. Each outputs a prediction and calibrated confidence. During inference, we follow the MDP

policy, compute confidence at each step via the appropriate substacker (or raw model output if no substacker exists), and exit when confidence exceeds a threshold. This reduces the substacker space from $2^k$ (exponential) to $\mathcal{O}(k)$ per trajectory (linear). The two components are complementary: trajectory discovery identifies common execution paths, while prefix meta-learning trains aggregators only for those paths, avoiding the combinatorial explosion of training substackers for all possible model subsets.

We implement two variants. LazyStack-Sub trains specialized substackers for each trajectory prefix. LazyStack-Mask trains a unified stacker that accepts all model slots, using zeros for unexecuted models and a binary mask indicating which models ran. Both operate purely on model probability outputs, requiring no architectural access.

Experiments across 8 datasets in 4 domains validate the approach. We evaluate on vision (CIFAR-100, ImageNet-1K), tabular (NSL-KDD), text classification (AG News, SST-5), and LLM routing (MMLU, ARC-Challenge, HEADLINES). On NSL-KDD, LazyStack achieves 38x speedup while retaining 97% of full ensemble accuracy, outpacing ABC (Kolawole et al., 2025) by 7x. On CIFAR-100, it retains 97% of full stacker accuracy (75.9% vs 78%) at 2.5x speedup. On LLM routing, it achieves 77% on MMLU and 88% on ARC-Challenge, surpassing FrugalGPT (Chen et al., 2024a) and RouteLLM (Ong et al., 2025). Across all settings, LazyStack averages 2 to 3 models per sample and does better on all black-box baselines on accuracy-latency Pareto frontiers.

**Contributions.** (1) *Progressive stacking*, a paradigm unifying cascading with learned aggregation at every execution step, made tractable by our finding that 95%+ of samples concentrate on 3–8 trajectories despite exponentially many possible paths; (2) *Joint trajectory discovery and prefix meta-learning*, where MDPs discover that optimal ordering contradicts cheapest-first heuristics, and prefix-trained substackers reduce aggregator count from $2^k$ to $\mathcal{O}(k)$ per trajectory; and (3) *Extensive black-box evaluation* across 8 datasets demonstrating up to $38\times$ speedup at 97%+ accuracy retention, with most samples requiring only 2–3 models regardless of ensemble size.

## 2. Related Work

Adaptive inference methods balance computational cost against prediction quality. We organize related work along four dimensions, focusing on methods operating in our black-box setting.

**Sequential Cascades.** Cascade classifiers (Viola & Jones, 2001) pioneered adaptive inference by sequencing models and terminating when confidence exceeds a threshold. We compare against three cascade variants. *Cost Cascade* (Lebovitz et al., 2023) orders models by cost and exits on single-model confidence. *FrugalGPT* (Chen et al., 2024a) trains DistilBERT (Sanh et al., 2019) scorers to predict success probabilities for cascade decisions over LLMs. *Agreement-Based Cascading (ABC)* (Kolawole et al., 2025) sequences heterogeneous models by complexity and stops when variance-based agreement is reached, aggregating via averaging. *Gatekeeper* (Rabanser et al., 2025) fine-tunes small models to output calibrated confidence for deferral; we compare on vision tasks where white-box access is available. These methods rely on fixed orderings or greedy exit rules. LazyStack instead learns both ordering and aggregation: an MDP discovers effective trajectories, and substackers aggregate predictions at each step. Recent work studies efficiency along complementary axes: Li et al. (2023) improve inference efficiency of a *homogeneous* deep ensemble

via shared early-exit layers inside a single white-box architecture, incompatible with our heterogeneous black-box setting; Nie et al. (2024) learn cascades *online* over data streams, complementary to our offline trajectory discovery; and Dekoninck et al. (2025) unify routing and cascading for LLMs, against which we compare directly (Table 26), where progressive stacking surpasses the single-model ceiling that routing is bounded by.

**Router-Based Selection.** Routers predict a single best model per input rather than executing sequentially. *RouteLLM* (Ong et al., 2025) trains routers on preference data to predict win rates between model pairs; we compare on LLM benchmarks. ThriftLLM (Huang et al., 2025) formulates selection as submodular maximization under cost budgets; we omit comparison as it targets one-shot selection without sequential fallback, a complementary setting. Routers minimize latency but forfeit fallback capability when predictions fail. LazyStack accumulates evidence sequentially, providing robustness through progressive aggregation.

**Mixture-of-Experts.** MoE methods (Jacobs et al., 1991; Shazeer et al., 2017; Lepikhin et al., 2020) use soft gating to weight expert contributions, requiring joint training of gates and experts. MoNE (Jain et al., 2024) routes tokens to nested sub-networks within vision transformers; MoD (Raposo et al., 2024) allocates compute at the token level within transformers. We omit comparison as these methods require white-box access for end-to-end training and operate within homogeneous architectures—assumptions incompatible with our heterogeneous black-box setting.

**Dynamic Ensemble Selection.** Classical DES methods select competent models per sample from trained pools (Cruz et al., 2018). KNORA (Ko et al., 2008) selects classifiers based on neighborhood accuracy; MetaDES (Cruz et al., 2015) extends this via meta-learning on multiple competence criteria. Hellsemble (Piwko et al., 2025) partitions data by difficulty, routing samples to specialized learners; we omit comparison as it requires coordinated training of base learners on difficulty-stratified data, incompatible with pre-trained or API-based models. LazyStack brings similar ideas to heterogeneous ensembles through MDP-based trajectory discovery and learned aggregation, requiring only probability outputs.

**Positioning LazyStack.** Cascades provide early exit but lack learned aggregation; routers minimize latency but forfeit fallback; MoE requires white-box access; DES relies on KNN estimation or coordinated training. LazyStack combines trajectory-based sequencing with learned aggregation over heterogeneous black-box models.

# 3. LazyStack

Efficient ensemble inference requires deciding which models to run and when to stop. Running all models guarantees accuracy but wastes compute on easy inputs; stopping too early risks incorrect predictions. We propose LazyStack, which learns both decisions from data: an MDP discovers effective model orderings, and trained aggregators determine when accumulated evidence is sufficient.

Like cascades, LazyStack executes models sequentially and exits early when confident. Unlike cascades, it aggregates *all* predictions seen so far rather than relying on single-model confidence, and it learns execution order rather than using fixed heuristics. The result: cascade-like efficiency when one model suffices, ensemble-quality predictions when aggregation helps.

Figure 2 illustrates the two-stage framework. In the offline phase, *Trajectory Discovery* (Section 3.1) formulates model selection as an MDP that trades off accuracy reward ($+\beta$ for correct predictions) against execution cost ($-\alpha \cdot c_j$ per model). Solving this MDP yields a policy $\pi^*$ and a small set of common trajectories $\mathcal{T}$ (typically 3–8 sequences covering 95%+ of samples). *Prefix Meta-Learning* (Section 3.2) then trains lightweight aggregators called *substackers* for every prefix of each trajectory, enabling calibrated confidence estimates from partial executions.

In the online phase, *Early Exit Inference* (Section 3.3) follows the learned policy: execute a model, aggregate all predictions via the substacker(or raw model output), and exit if confidence exceeds threshold $\theta$; otherwise continue. We provide two variants: LazyStack-Sub trains specialized substackers per trajectory prefix, while LazyStack-Mask trains a single universal substacker using learned masking.

We first describe trajectory discovery (Section 3.1), then prefix meta-learning (Section 3.2), and finally inference and implementation details (Section 3.3).

## 3.1. Trajectory Discovery

Given $N$ models with per-model costs $\{c_1, \ldots, c_N\}$, trajectory discovery identifies *which* models to execute and in *what order* (Figure 2, left). The output is a small set of trajectories $\mathcal{T} = \{\tau_1, \ldots, \tau_K\}$, each specifying an ordered model sequence. Model selection exhibits *delayed rewards*: executing a model incurs immediate cost, but the payoff (a confident prediction) may only emerge after several models run. Greedy stopping and ordering heuristics cannot capture this tradeoff, so we formulate trajectory discovery as a Markov Decision Process, which naturally handles sequential decisions where early actions affect later outcomes.

**MDP formulation.** We define the MDP tuple $(\mathcal{S}, \mathcal{A}, P, R, \gamma)$ with the objective of learning a policy $\pi^*$

that balances prediction accuracy (reward weight $\beta$) against execution cost (penalty weight $\alpha$) (Puterman, 1994).

*Actions.* At each step, the agent either executes a model or stops: $\mathcal{A} = \{\texttt{EXEC}(M_1), \ldots, \texttt{EXEC}(M_N)\} \cup \{\texttt{STOP}\}$. Executing $M_j$ incurs cost $c_j$ and yields a class probability distribution. Stopping terminates the episode yielding a trajectory of models executed so far.

*States.* Let $E \subseteq \{1, \ldots, N\}$ denote executed models. To keep the state space tractable, we summarize only the most recent model's output $p \in \mathbb{R}^C$: the predicted class $k = \arg\max_i p_i$ and discretized entropy $h \in \{1, \ldots, B\}$, where $H(p) = -\sum_{i=1}^{C} p_i \log p_i$ is binned into $B$ quantiles (we ablate $B$ in App. J). The full state is $s = (E, h, k)$, with initial state $s_0 = (\emptyset, B, \texttt{null})$ where $h = B$ (maximum entropy bin) reflects complete uncertainty before any model executes. This yields state space $O(2^N \cdot B \cdot C)$, though the reachable space is smaller since $E$ grows monotonically within each episode. The MDP discovers execution order; aggregation is handled separately by substackers (Section 3.2). We ablate alternative state representations in App. I.

*Transitions.* We estimate transitions empirically from validation set $\mathcal{D}_{\text{val}}$:

$$P(s' \mid s, M_j) = \frac{\sum_{x \in \mathcal{D}_{\text{val}}} \mathbf{1}[s(x) = s] \cdot \mathbf{1}[s'(x) = s' \text{ after } M_j]}{\sum_{x \in \mathcal{D}_{\text{val}}} \mathbf{1}[s(x) = s]}$$

(1)

For states with fewer than 10 samples, we back off to marginal distributions $P(h, k)$.

*Rewards.* Let $\hat{y}(s') = k'$ denote the predicted class in state $s'$ (i.e., the $\arg\max$ of the most recent model's output, used as a proxy for final prediction during MDP training) and $y$ the true label:

$$R(s, a, s') = \begin{cases} \beta \cdot \mathbf{1}[\hat{y}(s') = y] - \sum_{j \in E'} c_j & \text{if } a = \texttt{STOP} \\ -\alpha \cdot c_j & \text{if } a = \texttt{EXEC}(M_j) \end{cases}$$

(2)

Here $E'$ denotes the set of executed models in state $s'$. Each execution incurs penalty $\alpha \cdot c_j$; stopping yields $\beta$ if correct, minus total cost incurred along the trajectory.

**Policy optimization.** We solve for $\pi^*$ via value iteration (Sutton & Barto, 2018; Bellman, 1957):

$$V^*(s) = \max_{a \in \mathcal{A}(s)} \sum_{s'} P(s' \mid s, a) \left[ R(s, a, s') + \gamma V^*(s') \right] \quad (3)$$

with discount $\gamma \in (0, 1)$, iterating until convergence (change $< 10^{-4}$). The policy selects $\pi^*(s) = \arg\max_a \sum_{s'} P(s' \mid s, a)[R(s, a, s') + \gamma V^*(s')]$.

**Trajectory extraction.** We roll out $\pi^*$ on each $x \in \mathcal{D}_{\text{val}}$, recording executed model sequences as trajectories. We rank by coverage and retain top $K$ trajectories as $\mathcal{T}$. Our

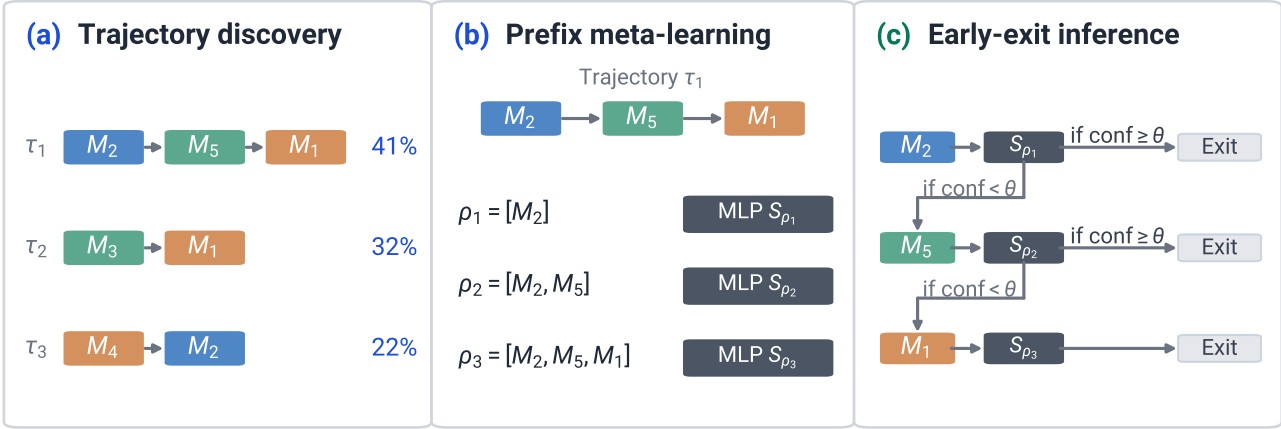

*Figure 2.* **LazyStack framework. (a)** Trajectory discovery formulates execution as an MDP over validation data and yields a small set of trajectories $\mathcal{T}$ that cover 95%+ of inputs. **(b)** Prefix meta-learning trains a substacker $S_\rho$ for every prefix $\rho$ of every trajectory; each substacker returns a calibrated $(\hat{y}, \mathrm{conf})$ pair from the predictions seen so far. **(c)** At test time we follow the policy: execute a model, aggregate via the substacker for the current prefix, and exit when $\mathrm{conf} \geq \theta$.

finding: Empirically, $K \leq 8$ trajectories suffice for high coverage (see Sec. 4). Uncovered samples fall back to the full ensemble. App. K analyzes the coverage-accuracy tradeoff.

### 3.2. Prefix Meta-Learning

Given top $K$ trajectories $\mathcal{T} = \{\tau_1, \ldots, \tau_K\}$ from the MDP, we train *substackers*: lightweight MLPs that combine predictions from model subsets into calibrated probability distributions (Figure 2, middle). The key idea is training a substacker for every *prefix* of each trajectory, not just complete sequences.

For a trajectory $\tau = [M_{j_1}, \ldots, M_{j_\ell}]$, we extract all prefixes: $[M_{j_1}]$, $[M_{j_1}, M_{j_2}]$, etc. The complete prefix set $\mathcal{P}$ is the union across all trajectories, with duplicates removed. Shared prefixes are common: trajectories $[M_2, M_5, M_1]$ and $[M_2, M_5, M_3]$ share prefixes $[M_2]$ and $[M_2, M_5]$, requiring only 4 substackers rather than 6. In practice, $K=6$ trajectories yield $|\mathcal{P}| \approx 12\text{--}18$ substackers.

Traditional cascades use only the *last* model's confidence to decide whether to continue. If $M_5$ is uncertain but $M_2$ was confident on the same class, a cascade continues unnecessarily. Substackers aggregate all predictions seen so far, leveraging accumulated evidence for earlier exit.

**LazyStack variants.** Figure 3 illustrates our two variants. Both take model probability outputs $p_j \in \mathbb{R}^C$ as input and produce calibrated predictions. LazyStack-Sub trains a specialized network $f_\rho^{\mathrm{custom}}$ for each prefix $\rho \in \mathcal{P}$, concatenating only the executed models' predictions. LazyStack-Mask trains a single shared network $f^{\mathrm{shared}}$ that accepts all $N$ model slots, using zeros for unexecuted models and a binary

mask $\mathbf{m} \in \{0, 1\}^N$ indicating which models ran. Formalization of the training for both variants appear in App. F.

A key challenge for LazyStack-Mask is that prefixes in $\mathcal{P}$ cover only a subset of possible model combinations. At test time, early exits or policy deviations may produce novel combinations unseen during training, leading to poorly calibrated confidence estimates. To address this, we augment training with random model subsets: for each sample, we generate $R=5$ additional masks where $m_j \sim \mathrm{Bernoulli}(f_j)$ independently, with $f_j$ proportional to how often $M_j$ appears in $\mathcal{T}$. This exposes $f^{\mathrm{shared}}$ to diverse input patterns, reducing expected calibration error by 15–20% on held-out combinations (App. L) and enabling more aggressive early exit.

**Variant tradeoffs.** LazyStack-Sub achieves higher accuracy through specialization, at the cost of managing $|\mathcal{P}|$ networks. LazyStack-Mask requires only one network, simplifying deployment. The mask variant can also achieve higher speedup because augmentation improves calibration, enabling earlier exit with reliable confidence estimates.

### 3.3. Early Exit Inference

At test time, we combine the learned policy $\pi^*$ with trained substackers (Figure 2, right). The procedure is simple: execute models according to $\pi^*$, aggregate predictions via substackers, and exit when confident.

**Confidence-based early exit.** After each model executes, we check whether to exit. Let $\rho_t$ denote the sequence of models executed so far at step $t$. If $\rho_t \in \mathcal{P}$, the substacker $S_{\rho_t}$ produces a calibrated probability distribution; otherwise we use the current model's output directly. We define con-

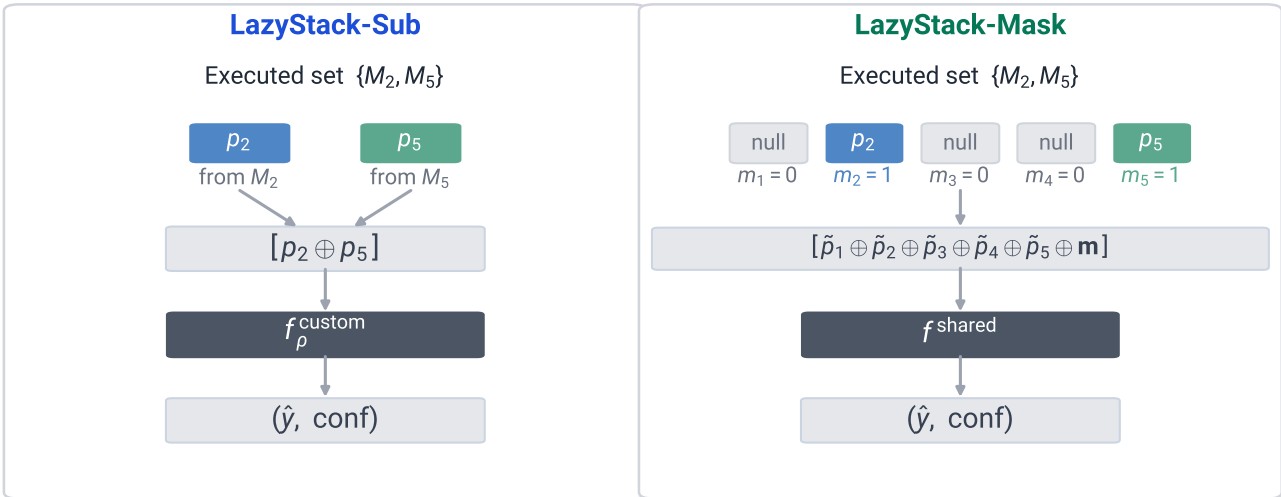

*Figure 3.* **Substacker variants.** Executed set is $\{M_2, M_5\}$ from a 5-model ensemble. **Left:** LazyStack-Sub concatenates only the executed models' predictions and feeds them to a prefix-specific MLP $f_\rho^{\mathrm{custom}}$. **Right:** LazyStack-Mask keeps a slot for every model, fills unexecuted slots with a null embedding ($\tilde{p}_i = p_i$ if $M_i$ ran, else null), appends the binary mask $\mathbf{m}$, and uses a single shared MLP $f^{\mathrm{shared}}$.

fidence as the maximum probability: $\mathrm{conf}(\mathbf{p}) = \max_i p_i$. When confidence exceeds threshold $\theta$, we exit and return prediction $\arg\max_i p_i$. Higher $\theta$ demands more certainty before stopping, preserving accuracy at the cost of speedup.

**Execution flow.** At every step we query $\pi^*(s)$ for an action. If $\pi^*$ returns $\mathrm{EXEC}(M_j)$, we run $M_j$, append it to the current execution sequence $\rho$, and check confidence: if $\rho \in \mathcal{P}$ we use $S_\rho$ to aggregate; otherwise we fall back to the model's own probability. We exit when $\mathrm{conf} \geq \theta$ or when $\pi^*$ returns STOP. LazyStack-Mask skips the $\rho \in \mathcal{P}$ check and always uses the single shared $S_{\mathrm{mask}}$. Full pseudocode for both variants is in App. G.

**Threshold calibration.** We calibrate $\theta$ on held-out validation data: compute full ensemble accuracy $\rho_{\mathrm{full}}$, set target retention (e.g., $\rho_{\mathrm{target}} = 0.97 \cdot \rho_{\mathrm{full}}$), sweep $\theta \in \{0.05, 0.10, \ldots, 0.95\}$, and select the highest $\theta$ achieving the target. This typically yields $\theta^* \in [0.25, 0.35]$. App. E provides details.

**Computational overhead.** Efficiency gains come from executing fewer models, not faster meta-learners. Substacker forward passes add under 1ms, policy lookup under 0.5ms via hash table, and prefix matching ($\rho \in \mathcal{P}$) under 0.1ms via prefix tree. App. U provides a full breakdown.

## 4. Experiments

We evaluate LazyStack across 8 datasets spanning vision, tabular, text, and LLM routing. Our focus is the *black-box* setting, where methods access models only through probability outputs.

*Table 1.* Datasets and ensembles used in evaluation. **Cost** is the ratio of the most expensive to the cheapest model latency (e.g., $65\times$ on NSL-KDD = LSTM 52ms / LogReg 0.8ms).

| Dataset | Task | Cls | Mod | Cost |
|---|---|---|---|---|
| *Vision* | | | | |
| CIFAR-100 (Krizhevsky, 2009) | Image cls. | 100 | 6 | 3.7× |
| ImageNet-1K (Russakovsky et al., 2015) | Image cls. | 1K | 8 | 11× |
| *Tabular* | | | | |
| NSL-KDD (Tavallaee et al., 2009) | Intrusion det. | 5 | 10 | 65× |
| *NLP* | | | | |
| AG News (Zhang et al., 2015) | Topic cls. | 4 | 5 | 4.2× |
| SST-5 (Socher et al., 2013) | Sentiment | 5 | 6 | 5.1× |
| *LLM Routing* | | | | |
| MMLU (Hendrycks et al., 2021) | QA | 4 | 5 | 17× |
| ARC-Challenge (Clark et al., 2018) | Sci. reasoning | 4 | 5 | 17× |
| HEADLINES (Sinha & Khandait, 2020) | News cls. | 3 | 5 | 17× |

### 4.1. Evaluation Setup

Next, we briefly include our datasets and baselines.

**Datasets and ensembles.** In Table 1, we detail our evaluation suite of eight datasets across four domains with diverse cost structures. App. A provides full model specifications for each ensemble and App. D details training procedures.

**Baselines.** We compare against methods that operate on model probability outputs:

• **ABC** (agreement-based stopping): Orders models by accuracy and exits when prediction variance falls below a threshold (Kolawole et al., 2025).

- **Cost Cascade** (confidence-based stopping): Orders models by cost and exits when single-model confidence exceeds a threshold. LLM Routing datasets only (Lebovitz et al., 2023).

- **FrugalGPT** (learned quality scoring): Trains a scorer to predict output quality for cascade decisions. LLM Routing datasets only. (Chen et al., 2024a).

- **RouteLLM** (single-model routing): Routes each query to exactly one model based on predicted win rate. LLM Routing datasets only. (Ong et al., 2025).

- **Gatekeeper** (confidence calibration): Fine-tunes small models to output calibrated confidence for deferral. Privileged baseline with white-box model access (Rabanser et al., 2025).

Additional white-box methods are discussed in App. V. Full baseline implementation details appear in App. C.

**Metrics.** We report accuracy and speedup. For non-LLM tasks, speedup is the ratio of full ensemble latency to average method latency. For LLM tasks, speedup is the ratio of always using the largest model (70B) to average per-query cost, following prior work (Chen et al., 2024a; Ong et al., 2025). All methods use thresholds to control the accuracy-speedup tradeoff; we sweep thresholds on validation data and report Pareto frontiers.

**Data splits and training.** Base models are *pre-trained*—LazyStack never retrains them. For LLM benchmarks we use off-the-shelf instruction-tuned models, so 100% of the benchmark data is available to LazyStack. For non-LLM benchmarks, base models are trained on the standard training split, while *all* LazyStack components are learned exclusively on a held-out validation split (an 80/20 train/validation partition). The validation portion serves three roles: MDP transition estimation, substacker training, and confidence-threshold calibration. For MMLU, this is train (99K) for the MDP and substackers, dev (1.5K) for calibration, and test (14K) for evaluation only. The test split is never seen during training; full details appear in App. A and App. D.

### 4.2. Results for LLM Routing Datasets

We begin with LLM datasets, where the 17× cost gap between small (7–8B) and large (70–72B) models makes routing decisions particularly impactful. App. B details dataset partitioning and probability extraction; App. R provides extended analysis. In Figure 4, we compare LazyStack against baselines on three datasets.

LazyStack-Sub achieves the highest accuracy on all three datasets: 77.2%±0.3 on MMLU (versus ABC 76.5%±0.4, FrugalGPT 75.8%±0.3), 88.2%±0.2 on ARC-Challenge (versus 87.5%±0.3, 86.8%±0.4), and 64.3%±0.4 on HEAD-

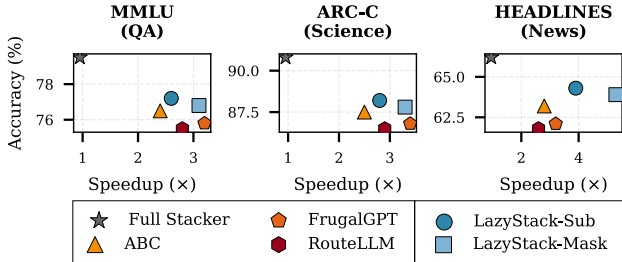

*Figure 4.* LLM routing Pareto comparison. LazyStack-Sub (blue) achieves highest accuracy among efficient methods, retaining 97%+ of full stacker at 2.6–3.9× speedup.

LINES (versus 63.2%±0.5, 62.1%±0.4). All intervals report standard deviation over 5 runs.

The MDP discovers non-trivial routing patterns. On MMLU, 62% of samples use only small models, 23% skip the medium tier to go small-then-large, and 15% use small-then-medium. The medium tier serves as a useful intermediate: confident enough to resolve queries that stump small models, but avoided when uncertainty suggests the large model is truly needed.

What do substackers learn on multiple-choice datasets where answer tokens (A, B, C, D) lack semantic meaning? They learn *confidence calibration*: which models produce reliable probabilities, how agreement correlates with correctness, and how to weight predictions from models with different strengths. On HEADLINES, a semantic classification task, substackers can additionally learn class-level patterns—correspondingly, we see larger gains there (1.1% over ABC versus 0.7% on MMLU).

### 4.3. Results for Vision, Text, and Tabular Datasets

The benefits of progressive stacking extend beyond LLM routing. In Figure 5, we compare LazyStack against baselines across vision, text, and tabular domains. The Full Stacker baseline, which executes all models on every sample, marks the accuracy upper bound at 1× speedup.

LazyStack consistently outperforms ABC across all five datasets. The gains are largest where cost heterogeneity is highest: on NSL-KDD (65× cost ratio), LazyStack-Sub achieves 76.8%±0.3 accuracy at 38±1.8× speedup versus ABC's 75.7%±0.4 at 5.1±0.3×—7× faster while also more accurate. The MDP routes easy samples through cheap models (averaging 2.2 models per sample), reserving expensive models for genuinely ambiguous cases. On text classification, LazyStack achieves similar accuracy improvements with 1.4–1.6× better speedup than ABC. Vision tasks show more modest gains (2.5–2.7× speedup) because many classes spread probability mass thin, requiring more models before confidence thresholds are reached. We additionally adapt FrugalGPT and RouteLLM to these non-LLM bench-

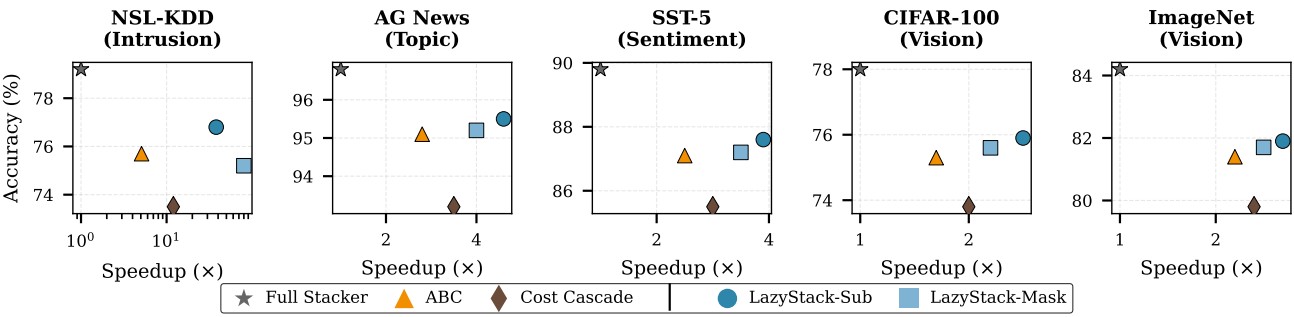

*Figure 5.* Pareto comparison on non-LLM datasets. Full Stacker (gray star) shows accuracy upper bound. LazyStack-Sub (blue) retains 97%+ of full stacker accuracy while achieving 2.5–38× speedup, consistently outperforming ABC (orange) across all datasets.

*Table 2.* Component ablation across domains (accuracy % / speedup). Both components contribute substantially across all four benchmarks; removing both collapses to near 1×.

| Variant | NSL-KDD | ImageNet | AG News | MMLU |
|---|---|---|---|---|
| Full LazyStack-Sub | **76.8 / 38×** | **82.4 / 2.7×** | **95.3 / 4.5×** | **77.2 / 2.6×** |
| w/o Trajectory Discovery | 76.2 / 29× | 82.0 / 2.2× | 95.1 / 3.4× | 76.9 / 2.1× |
| w/o Prefix Meta-Learning | 75.9 / 25× | 81.6 / 1.9× | 94.9 / 3.2× | 76.5 / 1.8× |
| w/o Both | 76.4 / 1.3× | 82.1 / 1.1× | 95.2 / 1.08× | 77.1 / 1.14× |

marks and re-run CIFAR-100 with a stronger SOTA ensemble; LazyStack-Sub dominates both baselines and retains ∼97% of the stronger ensemble (App. S).

### 4.4. Analysis

We analyze LazyStack's components through ablations and compare against a white-box baseline. **Ablations.** Table 2 ablates LazyStack's two components across four benchmarks spanning all domains. Each ablation replaces a learned module with a non-learned heuristic: *w/o Trajectory Discovery* replaces the MDP policy with a fixed cheapest-first ordering (substackers still trained for that sequence); *w/o Prefix Meta-Learning* keeps the MDP ordering but exits on raw single-model confidence; *w/o Both* combines cheapest-first ordering with raw confidence, reducing to a traditional cascade.

**Trajectory discovery enables non-obvious orderings.** Removing trajectory discovery reduces speedup by 24% on NSL-KDD (38× to 29×) and by 4–24% across the other domains. On NSL-KDD the MDP learns that LightGBM (2.1ms) outperforms cheaper Logistic Regression (0.8ms) as the first model—LightGBM's higher confidence enables earlier exit, more than compensating for its higher per-query cost. Greedy cost-based ordering misses this.

**Progressive aggregation improves exit calibration.** Removing prefix meta-learning reduces speedup by 28–34% across benchmarks. Single-model confidence is poorly calibrated; aggregating all predictions seen so far produces more reliable estimates that enable confident early exit.

**The components interact superadditively.** Removing both components collapses speedup to near 1× on every benchmark—far worse than either ablation alone. The fixed cascade (w/o both) uses cost-based ordering and runs models sequentially without ever reaching high confidence, forcing near-complete ensemble execution on most samples. Greedy ordering alone still identifies a few cost-saving trajectories, and single-model confidence alone still enables early exit when individual models are confident (though sometimes confidently wrong); the full system combines both into reliable, fast early exit.

**Better than privileged baselines.** While LazyStack operates under black-box constraints, we probe whether this costs significant performance by comparing against white-box methods. Gatekeeper (Rabanser et al., 2025) is a notable exception: it fine-tunes small models for calibrated deferral and supports heterogeneous model pairs, though only for vision tasks in non-LLM settings. On CIFAR-100 and ImageNet-1K, LazyStack-Sub matches or exceeds Gatekeeper's best 2-model configuration on both accuracy and speedup despite requiring only probability outputs. On ImageNet-1K, LazyStack achieves 82.4% accuracy at 2.7× speedup, matching Gatekeeper's best speedup while exceeding its accuracy by 0.8%. Full results appear in App. V.

**Qualitative routing behavior.** Figure 6 illustrates a pattern that recurs across all eight benchmarks (full case studies in App. T). On confusable inputs the cheapest model is often confidently wrong—on the ImageNet *tiger*, the first model predicts *leopard*—and Cost-Cascade exits on this error, whereas LazyStack's substacker corrects it after a single additional model and exits, matching Stacking's accuracy at a fraction of its cost. The discovered orderings are themselves interpretable: on NSL-KDD the policy opens with LightGBM rather than the cheaper Logistic Regression, because LightGBM's better-calibrated confidence resolves most traffic in one step; on the LLM benchmarks it learns a "skip-medium" policy, routing 13–18% of queries directly from a small model to a large one when the intermediate tier adds no information. Easy inputs exit after one model while

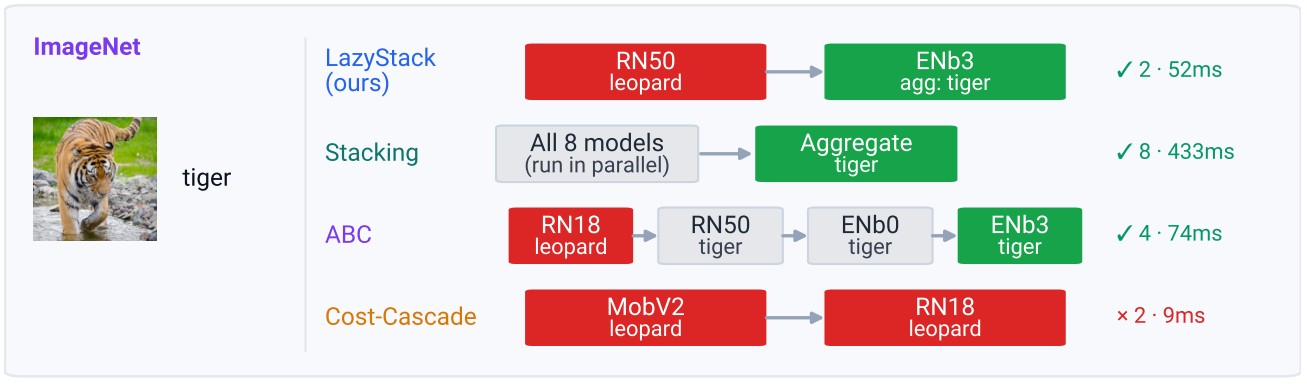

*Figure 6.* **What the four methods actually do on a single sample.** On the ImageNet tiger, Cost-Cascade exits on a wrong single-model guess (9 ms). ABC runs four models before agreeing on the correct answer (74 ms). Stacking runs the full 8-model ensemble (433 ms). LazyStack's substacker corrects the first-model error after just one extra call (52 ms)—matching Stacking's accuracy at 12× lower cost. App. T extends this to all eight benchmarks.

genuinely ambiguous ones (e.g., visually similar CIFAR-100 classes) traverse longer prefixes, so cost scales with sample difficulty rather than with ensemble size.

**Additional analyses.** We provide extensive supplementary experiments organized by theme. Prefix-meta-learning alternatives, NSL-KDD F1/AUROC, a stronger SOTA ensemble on CIFAR-100, FrugalGPT/RouteLLM comparisons, decoupled-vs-joint training, and a head-to-head with cascade-routing (Dekoninck et al., 2025) appear in App. S; qualitative case studies are in App. T.

**Two properties drive efficiency gains.** First, trajectory concentration: despite $2^k$ possible paths, 95%+ of samples follow just 3 to 8 trajectories (App. K). Second, sublinear model usage: as ensembles grow from 3 to 10 models on NSL-KDD, speedup increases from 3.2× to 36.8× while average models executed grows only from 1.7 to 2.2 (App. M). Function approximation extends this to 30+ models (App. N).

**The MDP learns a skip-medium pattern on LLM tasks.** On LLM datasets, 62 to 71% of queries resolve with small models alone. The MDP also routes 13 to 18% of samples directly from a small to a large model, bypassing the 32B tier entirely when intermediate uncertainty provides no signal (App. R).

**Routing adapts to sample complexity.** Difficult samples consume more models while easy ones exit early (App. O); Cost-Cascade outperforms LazyStack on only 3.5% of samples (App. P).

*Implementation details.* Substacker forward passes and policy lookup add 7 to 26% overhead depending on base model costs (App. U). LazyStack-Sub achieves 0.2 to 0.3% higher accuracy through specialization, while LazyStack-Mask offers simpler deployment (App. H). Accuracy varies by

±0.3% across hyperparameter ranges (App. L). Controlling for cost heterogeneity, LazyStack achieves 1.3 to 7.5× higher normalized efficiency than ABC (App. Q).

## 5. Conclusion

We introduced LazyStack, a framework unifying stacking and cascading for cost-efficient ensemble inference. Learned substackers enable confident early exits by aggregating accumulated evidence, while MDP-based trajectory discovery identifies effective model orderings—including non-obvious ones like preferring a moderately expensive model over the cheapest when its higher confidence enables earlier overall exit. These components interact superadditively: across 8 datasets, LazyStack achieves 2.5–38× speedup at 97%+ accuracy retention, averaging just 2–3 models per sample. The framework requires only probability outputs, adding minimal overhead while generalizing across vision, text, tabular, and LLM domains. App. W provides method selection guidelines, and App. X offers a complete deployment guide.

## Acknowledgments

The research reported herein was supported by Army Research Office (ARO) award W911NF-24-2-0114. The views and opinions presented herein are those of the authors and do not necessarily represent the views of ARO or its components. Ashwin was also supported in part by HPI@UCI.

## Impact Statement

This work reduces the computational cost of ensemble inference. The primary societal benefits include reduced energy consumption and carbon emissions from ML deployment, and democratized access to ensemble-quality predictions

for organizations with limited compute budgets. We do not foresee negative consequences specific to this work beyond those inherent to improving ML systems generally; LazyStack makes existing methods more efficient rather than introducing new capabilities.

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

# Appendix Overview

This appendix provides supplementary material organized into five groups:

**Setup & Reproducibility** (App. A–App. D)

- App. A. **Ensemble Compositions** – Model specifications, latencies, and standalone accuracies
- App. B. **LLM Experiment Setup** – Dataset partitioning, probability extraction, cost measurement
- App. C. **Baseline Details** – Implementation details for all compared methods
- App. D. **Training Details** – Base model and LazyStack training procedures

**Method Details** (App. E–App. G)

- App. E. **Threshold Calibration** – Per-method threshold mechanisms
- App. F. **Substacker Training** – Formalization of Substacker Training
- App. G. **Inference Algorithms** – Complete algorithm for LazyStack-Mask

**Ablations & Sensitivity** (App. H–App. L)

- App. H. **Additional Ablations** – Threshold sensitivity, variant comparison, MDP components
- App. I. **MDP State Representation** – Ablation on state history depth
- App. J. **Entropy Discretization** – Effect of bin count $B$
- App. K. **Trajectory Coverage Threshold** – Coverage vs. accuracy tradeoff
- App. L. **Hyperparameter Sensitivity** – Sensitivity analysis across all parameters

**Extended Results** (App. M–App. R)

- App. M. **Scalability Results** – Performance vs. ensemble size
- App. N. **Scalability with Function Approximation** – Extending to 30+ models
- App. O. **Per-Class Analysis** – Performance breakdown by class difficulty
- App. P. **Failure Case Analysis** – When and why LazyStack underperforms
- App. Q. **Normalized Efficiency** – Controlling for cost heterogeneity
- App. R. **Extended LLM Analysis** – Routing patterns, per-subject breakdown, calibration
- App. S. **Additional Experiments** – Prefix-meta-learning alternatives, F1/AUROC, SOTA ensemble, Frugal-GPT/RouteLLM, joint training, cascade-routing
- App. T. **Qualitative Analysis** – Case-by-case routing across datasets vs. baselines

**Practical Guidance** (App. U–App. X)

- App. U. **Computational Overhead** – Inference time breakdown
- App. V. **White-Box Method Applicability** – Why MoNE, Model Soup, Gatekeeper don't generalize
- App. W. **Discussion** – Method selection guidelines, limitations
- App. X. **Practitioner's Guide** – Recommended defaults, deployment checklist

# A. Ensemble Compositions

This section provides full model specifications for each ensemble, as referenced in Sec. 4.1.

Table 3 and Table 4 list the models in each ensemble, ordered by inference cost. Vision ensembles span convolutional (MobileNetV2 (Sandler et al., 2018), ResNet (He et al., 2016), EfficientNet (Tan & Le, 2019)), attention-based (ViT-B/16 (Dosovitskiy et al., 2021)), and hybrid (ConvNeXt (Liu et al., 2022)) architectures; extending to other designs such as pixel-level transformers (Nguyen et al., 2025) requires no changes to LazyStack, as it operates solely on probability outputs.

*Table 3.* Vision ensemble compositions, ordered by cost (cheapest first).

| Dataset | Model | Latency (ms) | Standalone Acc. |
|---|---|---|---|
| CIFAR-100 | MobileNetV2 (Sandler et al., 2018) | 8.2 | 71.2% |
| | ResNet-18 (He et al., 2016) | 12.4 | 73.8% |
| | EfficientNet-B0 (Tan & Le, 2019) | 15.1 | 74.5% |
| | ResNet-50 | 21.3 | 76.2% |
| | EfficientNet-B4 | 28.7 | 77.1% |
| | ViT-B/16 (Dosovitskiy et al., 2021) | 30.5 | 76.8% |
| ImageNet-1K | MobileNetV2 | 12.1 | 72.0% |
| | EfficientNet-B0 | 18.4 | 77.1% |
| | ResNet-50 | 25.8 | 76.1% |
| | EfficientNet-B2 | 32.6 | 80.1% |
| | ResNet-101 | 45.2 | 77.4% |
| | EfficientNet-B4 | 68.3 | 82.9% |
| | ViT-B/16 | 98.7 | 81.8% |
| | ConvNeXt-B (Liu et al., 2022) | 132.4 | 83.8% |

# B. LLM Experiment Setup

This section details dataset partitioning and probability extraction for LLM experiments, as referenced in Sec. 4.2.

**Dataset Partitioning** LazyStack requires training data for MDP trajectory discovery and stacker training. Since standard LLM benchmarks are designed for zero-shot or few-shot evaluation, we partition available data carefully to avoid contamination.

**MMLU.** We use the auxiliary training set (99K samples across 57 subjects) for trajectory discovery and stacker training. The development set (1.5K samples) serves as our validation set for hyperparameter selection and confidence threshold calibration. We report results on the standard test set (14K samples). This partitioning follows the intended use of MMLU's auxiliary data, which was released specifically to enable training-based approaches.

**ARC-Challenge.** The ARC-Challenge training set contains only 1.1K samples—insufficient for reliable MDP learning. We augment this with ARC-Easy training data (2.3K samples), giving us 3.4K total training samples. We use the ARC-Challenge development set (299 samples) for validation and report results on the standard test set (2.6K samples). While ARC-Easy questions are simpler, they help the MDP learn basic routing patterns that transfer to harder questions.

**Probability Extraction** Both MMLU and ARC-Challenge are 4-way multiple choice tasks. For each LLM, we extract log-probabilities over answer tokens (A, B, C, D) and convert to a probability distribution via softmax. This gives us a 4-dimensional probability vector per model that can be aggregated by the stacker, analogous to class probabilities in image classification.

For instruction-tuned models, we use the prompt format recommended by each model's documentation. We found that probability calibration varies across models. For instance, Qwen models tend to be overconfident while LLaMA models are better calibrated. However, the stacker learns to account for these differences during training.

**Cost Measurement** We measure per-query latency using vLLM with batch size 1 on a single A100 GPU. Costs include tokenization, forward pass, and probability extraction. The 7-8B models average 45-52ms per query; Qwen2.5-32B averages 175-185ms; and 70-72B models average 780-890ms. These costs span roughly $17\times$ from smallest to largest, creating three

*Table 4.* Tabular and text ensemble compositions, ordered by cost (cheapest first).

| Dataset | Model | Latency (ms) | Standalone Acc. |
|---|---|---|---|
| NSL-KDD | Logistic Regression | 0.8 | 68.2% |
| | Decision Tree | 1.2 | 71.5% |
| | LightGBM (Ke et al., 2017) | 2.1 | 74.8% |
| | Random Forest | 3.4 | 73.2% |
| | CatBoost (Prokhorenkova et al., 2018) | 4.8 | 75.1% |
| | XGBoost (Chen & Guestrin, 2016) | 5.2 | 75.4% |
| | MLP | 8.7 | 72.8% |
| | SVM (RBF) | 18.3 | 71.9% |
| | 1D-CNN | 32.1 | 74.2% |
| | LSTM | 52.0 | 73.8% |
| AG News | DistilBERT (Sanh et al., 2019) | 8.2 | 93.1% |
| | BERT-base (Devlin et al., 2019) | 15.4 | 94.2% |
| | RoBERTa-base (Liu et al., 2019) | 16.1 | 94.5% |
| | ALBERT-large (Lan et al., 2020) | 24.8 | 94.1% |
| | RoBERTa-large | 34.6 | 95.2% |
| SST-5 | DistilBERT | 8.2 | 48.2% |
| | BERT-base | 15.4 | 52.1% |
| | RoBERTa-base | 16.1 | 53.8% |
| | ALBERT-large | 24.8 | 51.9% |
| | DeBERTa-base (He et al., 2021) | 28.3 | 54.2% |
| | RoBERTa-large | 41.8 | 55.6% |
| LLM Routing | Qwen2.5-7B-Instruct (Qwen Team, 2024) | 48 | varies |
| | LLaMA-3.1-8B-Instruct (Grattafiori et al., 2024) | 52 | varies |
| | Qwen2.5-32B-Instruct | 180 | varies |
| | LLaMA-3.1-70B-Instruct | 820 | varies |
| | Qwen2.5-72B-Instruct | 890 | varies |

distinct cost tiers that enable meaningful trajectory discovery: a system that routes most queries through small models with selective escalation achieves substantial cost savings.

## C. Baseline Details

This section provides implementation details for all baselines compared in Sec. 4.1.

**Full Stacker**   Our accuracy upper bound. We train a two-layer MLP (hidden dimensions 128, 64) on concatenated probability vectors from all base models. Training uses cross-entropy loss with Adam optimizer (learning rate $10^{-3}$), early stopping based on validation accuracy. This baseline executes all models for every sample.

**ABC (Agreement-Based Cascading)**   We implement ABC following Kolawole et al. (2025). Models are ordered by standalone accuracy (highest first). At each step, we compute the variance of predicted probabilities across executed models; if variance falls below a learned threshold, we stop and return the majority prediction. We tune the variance threshold on validation data to match Full Stacker accuracy within 0.5%.

**RouteLLM**   We implement RouteLLM following Ong et al. (2025). A BERT-based classifier is trained to predict, for each query, whether the small model (7-8B) or large model (70-72B) will answer correctly. At inference, queries are routed to exactly one model based on the classifier's prediction. We train the router on our training split and tune the decision threshold on validation data. Note that RouteLLM's binary routing cannot leverage the medium tier; we include it as a baseline representing the router-based paradigm.

**FrugalGPT**   We implement FrugalGPT following Chen et al. (2024a). The key innovation is a learned quality judge—a small model trained to predict whether an LLM's output is acceptable without access to ground truth. Models are cascaded in cost order (cheapest first). After each model, the judger scores the output; if the score exceeds a threshold, we return the answer, otherwise we continue to the next model. Unlike LazyStack, FrugalGPT does not aggregate predictions—it discards

earlier outputs when escalating. The judger is trained on a held-out portion of our training split using the approach described in the original paper.

**Fixed Cascade**  Models ordered by standalone accuracy (highest first). We apply a calibrated confidence threshold: if max probability exceeds $\theta$, stop and return the prediction. We tune $\theta$ on validation data.

**Cost Cascade**  Models ordered by cost (cheapest first), same confidence-based stopping as Fixed Cascade.

## D. Training Details

This section details training procedures for base models and LazyStack components, supporting Section 3.1 and Section 3.2.

### D.1. Base Model Training

**Vision (CIFAR-100, ImageNet-1K)**  All models trained using SGD with momentum 0.9, weight decay $10^{-4}$, cosine learning rate schedule. Initial learning rate 0.1 for ResNets, 0.01 for EfficientNets and ViT. Batch size 128 (CIFAR-100) or 256 (ImageNet). Standard augmentation: random crop, horizontal flip, color jitter. Training for 200 epochs (CIFAR-100) or 90 epochs (ImageNet).

**Tabular (NSL-KDD)**  Classical models (Logistic Regression, Decision Tree, Random Forest, SVM) use scikit-learn defaults with hyperparameter tuning via 5-fold cross-validation. Gradient boosting models (XGBoost, LightGBM, CatBoost) use 100–500 trees with early stopping based on validation loss. Neural models (MLP, 1D-CNN, LSTM) use Adam optimizer with learning rate $10^{-3}$, batch size 64, trained for 50 epochs with early stopping.

**Text (AG News, HEADLINES)**  All transformer models fine-tuned from HuggingFace pretrained checkpoints. AdamW optimizer with learning rate $2 \times 10^{-5}$, linear warmup for 10% of training steps, batch size 32. Maximum sequence length 128 (AG News) or 64 (HEADLINES). Training for 3 epochs.

**LLM (MMLU, ARC-Challenge)**  We use instruction-tuned variants without additional fine-tuning: Qwen2.5-7B-Instruct, Qwen2.5-32B-Instruct, Qwen2.5-72B-Instruct, LLaMA-3.1-8B-Instruct, and LLaMA-3.1-70B-Instruct. Inference uses vLLM with temperature 0 (greedy decoding) for deterministic probability extraction. The five models span three cost tiers, enabling the MDP to discover non-trivial routing patterns.

### D.2. LazyStack Training

**MDP Trajectory Discovery**  We run value iteration with discount factor $\gamma = 0.95$, convergence threshold $10^{-4}$. Reward weights: $\alpha = 0.2$ (cost penalty), $\beta = 10.0$ (correctness reward). Entropy discretization uses $B = 5$ bins computed via quantiles on validation predictions. Hierarchical smoothing activates when a state has fewer than 10 validation samples.

Value iteration typically converges in 50–100 iterations (under 1 minute on CPU). We then simulate the learned policy on all validation samples, recording execution traces. Trajectories covering at least 2% of validation samples are retained; this yields 3–8 trajectories per dataset.

**Stacker Training**  substackers are two-layer MLPs (hidden dimensions 64, 32) with ReLU activations and dropout 0.3. Training uses Adam optimizer with learning rate $10^{-3}$, batch size 64, early stopping with patience 10 epochs. Each substacker trains in under 30 seconds; total stacker training takes 3–5 minutes.

The Mask stacker has hidden dimensions 128, 64 to handle the larger input space (N model predictions plus N mask indicators). We augment training data with $R = 5$ random masks per sample, where mask probabilities are proportional to model frequency in discovered trajectories. Mask stacker training takes 5–10 minutes.

**Confidence Threshold Calibration**  We sweep $\theta \in \{0.05, 0.10, \ldots, 0.95\}$ on validation data, measuring accuracy and speedup at each threshold. We select $\theta^* = \max\{\theta : \text{accuracy} \geq 0.97 \times \text{Full Stacker accuracy}\}$. This typically yields $\theta^* \in [0.25, 0.35]$.

**Mask Training Data Generation.** Algorithm 1 details the training data construction for LazyStack-Mask. We generate examples from two sources: trajectory prefixes and random mask augmentation.

For trajectory prefixes, we iterate over each trajectory $\tau_i \in \mathcal{T}$ and each prefix length $k \in \{1, \ldots, |\tau_i|\}$. For each sample $x \in \mathcal{D}_{\text{train}}$, we execute the first $k$ models in $\tau_i$, construct the corresponding mask $\mathbf{m}$, and pair the input with label $y(x)$. With $K$ trajectories averaging $\bar{\ell} \approx 3.5$ models, this yields roughly $3.5K \times |\mathcal{D}_{\text{train}}|$ examples.

For random mask augmentation, we sample $R = 5$ random masks per sample. Each model $M_j$ is included independently with probability proportional to its frequency across all trajectories in $\mathcal{T}$. This adds $5 \times |\mathcal{D}_{\text{train}}|$ examples.

The combined dataset is larger than $\mathcal{D}_{\text{train}}$ but training remains fast (under 10 minutes) since $S_{\text{mask}}$ operates on low-dimensional probability vectors rather than raw inputs.

---

**Algorithm 1** Training Data Generation for LazyStack-Mask

---

**Input:** Training set $\mathcal{D}_{\text{train}}$, trajectories $\mathcal{T}$, augmentation factor $R$
**Output:** Augmented dataset $\mathcal{D}_{\text{aug}}$
$\mathcal{D}_{\text{aug}} \leftarrow \emptyset$ // Trajectory prefix examples
**foreach** *trajectory $\tau_i \in \mathcal{T}$* **do**
  **for** $k = 1$ **to** $|\tau_i|$ **do**
    **foreach** *sample $x \in \mathcal{D}_{train}$* **do**
      $\mathbf{m} \leftarrow \mathbf{0}^N$ **for** $\ell = 1$ **to** $k$ **do**
        $j \leftarrow \tau_i[\ell]$   $\mathbf{m}[j] \leftarrow 1$,    $\tilde{p}_j \leftarrow M_j(x)$
      Fill remaining $\tilde{p}_j \leftarrow 0$ where $\mathbf{m}[j] = 0$   $\mathcal{D}_{\text{aug}} \leftarrow \mathcal{D}_{\text{aug}} \cup \{([\tilde{\mathbf{p}}, \mathbf{m}], y(x))\}$

// Random mask augmentation
Compute $\text{freq}(M_j) =$ fraction of trajectories containing $M_j$   **foreach** *sample $x \in \mathcal{D}_{train}$* **do**
  **for** $r = 1$ **to** $R$ **do**
    **foreach** *model $M_j$* **do**
      $\mathbf{m}[j] \sim \text{Bernoulli}(\text{freq}(M_j))$   $\tilde{p}_j \leftarrow M_j(x)$ if $\mathbf{m}[j] = 1$ else $\mathbf{e}_{\text{null}}$
    **if** $\sum_j \mathbf{m}[j] > 0$ **then** $\mathcal{D}_{\text{aug}} \leftarrow \mathcal{D}_{\text{aug}} \cup \{([\tilde{\mathbf{p}}, \mathbf{m}], y(x))\}$
**return** $\mathcal{D}_{aug}$

---

### D.3. Computational Requirements

Table 5 reports the training-time breakdown.

| Stage | Time | Hardware |
|---|---|---|
| Base model training | 2–48 hours | 1 GPU |
| Validation prediction | 10–60 min | 1 GPU |
| MDP trajectory discovery | <1 min | CPU |
| Stacker training (Sub) | 3–5 min | CPU |
| Stacker training (Mask) | 5–10 min | CPU |
| Threshold calibration | 5–10 min | 1 GPU |
| **Total LazyStack overhead** | 15–75 min | — |

*Table 5.* Training time breakdown. LazyStack adds 15–75 minutes on top of base model training.

LazyStack's training overhead is modest. The dominant cost is computing validation predictions from all base models (10–60 minutes depending on dataset size and model count). MDP solving and stacker training are fast (under 15 minutes total on CPU). This makes LazyStack practical to deploy: once base models are trained, LazyStack can be configured in under an hour.

## E. Threshold Calibration Details

This section details threshold calibration mechanisms for all methods, as referenced in Section 3.3.

**LazyStack.** We threshold on the maximum probability of the stacker's output distribution. Given stacker output $\mathbf{p} \in \Delta^C$, we stop if $\max_c p_c > \theta$. The threshold $\theta$ is calibrated on validation data to achieve a target accuracy level (we use 97% of ensemble accuracy as the default).

**FrugalGPT.** FrugalGPT trains a generation scoring function $g(q, a) \to [0, 1]$ that estimates the reliability of an LLM's response $a$ to query $q$. If $g(q, a) < \tau_k$ for model $k$, the cascade escalates to model $k + 1$. Following Chen et al. (2024a), we jointly optimize the model sequence and per-model thresholds $\{\tau_k\}$ on validation data under a cost budget.

**RouteLLM.** RouteLLM predicts a "strong model win rate" $w(q) \in [0, 1]$ for each query $q$—the probability that the strong model would produce a better response than the weak model. If $w(q) > \tau$, the query routes to the strong model; otherwise to the weak model. Unlike cascade methods, RouteLLM makes a single routing decision without sequential execution. Following Ong et al. (2025), we calibrate $\tau$ to achieve a target percentage of queries routed to the strong model.

**ABC.** ABC (Kolawole et al., 2025) stops when $k$ consecutive models agree on the predicted class. We sweep $k \in \{2, 3, 4\}$ and the confidence threshold for agreement.

## F. Substacker Training Details

This section provides the formalization of the LazyStack variants in Section 3.2.

**LazyStack-Sub training.** For each prefix $\rho \in \mathcal{P}$, let $\mathcal{M}(\rho)$ denote the models in $\rho$. We construct dataset $\mathcal{D}^\rho_{\text{train}}$ where each $(x, y) \in \mathcal{D}_{\text{train}}$ yields the pair:

$$\left( \bigoplus_{M_j \in \mathcal{M}(\rho)} p_j(x), \; y \right) \tag{4}$$

The substacker $S_\rho$ is a two-layer MLP trained on $\mathcal{D}^\rho_{\text{train}}$ with cross-entropy loss.

**LazyStack-Mask training.** We construct $\mathcal{D}^{\text{mask}}_{\text{train}}$ from two sources. First, for each $\rho \in \mathcal{P}$ and $(x, y) \in \mathcal{D}_{\text{train}}$, we create:

$$\left( \left( \bigoplus_{j=1}^{N} \tilde{p}_j(x) \right) \oplus \mathbf{m}, \; y \right) \tag{5}$$

where $\tilde{p}_j(x) = p_j(x)$ if $M_j \in \mathcal{M}(\rho)$, and $\tilde{p}_j(x) = \mathbf{0} \in \mathbb{R}^C$ otherwise. The mask $\mathbf{m} \in \{0, 1\}^N$ indicates executed models: $m_j = 1$ iff $M_j \in \mathcal{M}(\rho)$. This yields $|\mathcal{P}| \times |\mathcal{D}_{\text{train}}|$ examples from trajectory prefixes.

Second, we add $R \times |\mathcal{D}_{\text{train}}|$ augmented examples using random masks as described in Section 3.2. The substacker $S_{\text{mask}}$ is a two-layer MLP trained on the combined dataset with cross-entropy loss.

## G. Inference Algorithms

This section provides the complete inference algorithms for both LazyStack-Sub and LazyStack-Mask, as referenced in Section 3.3.

LazyStack-Sub (Algorithm 2) checks at each step whether the current execution sequence $\rho$ matches a learned prefix and, if so, aggregates via $S_\rho$; otherwise it falls back to the executed model's own probabilities. LazyStack-Mask (Algorithm 3) drops the prefix-matching step: it maintains a slot vector $\tilde{\mathbf{p}}$ and a binary mask $\mathbf{m}$ and always aggregates via the universal $S_{\text{mask}}$.

The key difference from LazyStack-Sub is simplicity: no trajectory matching is needed, and confidence is always computed via $S_{\text{mask}}$ rather than falling back to single-model confidence. This makes LazyStack-Mask easier to deploy but requires that the mask stacker generalize well to arbitrary model combinations, which is why random mask augmentation during training is critical.

---

**Algorithm 2** Progressive inference, LazyStack-Sub

---

**Input:** Sample $x$, policy $\pi^*$, prefix set $\mathcal{P}$, substackers $\{S_\rho\}_{\rho \in \mathcal{P}}$, threshold $\theta$
**Output:** Prediction $\hat{y}$, cost $c$
$s \leftarrow s_0,\ \rho \leftarrow [\,],\ c \leftarrow 0$
**while** $s \neq s_{\text{terminal}}$ **do**
    $a \leftarrow \pi^*(s)$  **if** $a = STOP$ **then break**
    $j \leftarrow$ model index from $a = \texttt{EXEC}(M_j)$  $p_j \leftarrow M_j(x)$  $\rho \leftarrow \rho \oplus [M_j]$  $c \leftarrow c + c_j$
    **if** $\rho \in \mathcal{P}$ **then**
        $\mathbf{p} \leftarrow S_\rho\left(\bigoplus_{M_i \in \mathcal{M}(\rho)} p_i\right)$  **if** $\text{conf}(\mathbf{p}) > \theta$ **then**  **return** $\arg\max(\mathbf{p})$, $c$
    **else**
        **if** $\text{conf}(p_j) > \theta$ **then**  **return** $\arg\max(p_j)$, $c$
    Update $s$ with entropy of $p_j$ and predicted class
**return** $\arg\max(p_j)$, $c$                  `// fallback: full stacker`

---

**Algorithm 3** Progressive inference, LazyStack-Mask

---

**Input:** Sample $x$, policy $\pi^*$, threshold $\theta$
**Output:** Prediction $\hat{y}$, cost $c$
$s \leftarrow s_0,\quad \tilde{\mathbf{p}} \leftarrow [\mathbf{0}, \ldots, \mathbf{0}],\quad \mathbf{m} \leftarrow \mathbf{0}^N,\quad c \leftarrow 0$  **while** $s$ *not terminal* **do**
    $a \leftarrow \pi^*(s)$  **if** $a = STOP$ **then break**
    $j \leftarrow$ model index from $a = \texttt{EXEC}(M_j)$  $\tilde{\mathbf{p}}[j] \leftarrow M_j(x),\quad \mathbf{m}[j] \leftarrow 1,\quad c \leftarrow c + c_j$  $\mathbf{probs} \leftarrow \text{softmax}(S_{\text{mask}}([\tilde{\mathbf{p}}, \mathbf{m}]))$
    **if** $\max(\mathbf{probs}) > \theta$ **then return** $\arg\max(\mathbf{probs})$, $c$
    Update $s$ with entropy of $\tilde{\mathbf{p}}[j]$ and predicted class
**return** $\arg\max(\mathbf{probs})$, $c$                  `// full ensemble`

---

# H. Additional Ablation Studies

This section provides additional ablations beyond those in Sec. 4.4, including threshold sensitivity, variant comparison, and MDP component analysis.

## H.1. Confidence Threshold Sensitivity

The confidence threshold $\theta$ controls the accuracy-efficiency tradeoff. Table 6 shows results across threshold values on CIFAR-100 and NSL-KDD.

| | **CIFAR-100** | | **NSL-KDD** | |
|---|---|---|---|---|
| **Threshold $\theta$** | Acc | Speedup | Acc | Speedup |
| 0.20 (loose) | 74.89 | 4.12× | 73.89 | 52.1× |
| 0.25 | 75.24 | 3.78× | 74.52 | 44.3× |
| 0.30 (default) | 75.71 | 2.40× | 75.21 | 36.8× |
| 0.35 | 75.85 | 2.12× | 75.58 | 30.2× |
| 0.40 (strict) | 75.93 | 1.89× | 75.82 | 24.5× |

*Table 6.* Confidence threshold sensitivity. Looser thresholds trade accuracy for speedup. The default $\theta = 0.30$ balances both objectives.

The tradeoff is smooth and predictable. On NSL-KDD, relaxing $\theta$ from 0.30 to 0.20 improves speedup by 42% (36.8× to 52.1×) at a cost of 1.3% accuracy (75.21% to 73.89%). Practitioners can tune $\theta$ based on their accuracy requirements.

Interestingly, the threshold sensitivity differs across datasets. CIFAR-100 shows less sensitivity (accuracy ranges 74.89–75.93%, a 1.04% span) compared to NSL-KDD (73.89–75.82%, a 1.93% span). This reflects the difficulty distribution: CIFAR-100's many-class structure means even confident predictions are often wrong, so threshold changes have less impact.

## H.2. Variant Comparison: LazyStack-Sub vs LazyStack-Mask

Table 7 compares our two variants across all datasets.

| Dataset | LazyStack-Sub | | | LazyStack-Mask | | |
|---|---|---|---|---|---|---|
| | Acc | Speed | AME | Acc | Speed | AME |
| CIFAR-100 | 75.71 | 2.40× | 2.71 | 75.48 | 2.18× | 2.91 |
| ImageNet-1K | 81.92 | 2.74× | 3.12 | 81.78 | 2.51× | 3.34 |
| NSL-KDD | 75.21 | 36.8× | 2.21 | 74.89 | 79.6× | 2.03 |
| AG News | 95.32 | 4.50× | 2.14 | 95.08 | 3.89× | 2.32 |
| Headlines | 87.38 | 3.73× | 2.08 | 87.05 | 3.45× | 2.21 |

*Table 7.* Variant comparison. Sub achieves higher accuracy through specialized stackers; Mask offers deployment simplicity and can achieve extreme speedups (79.6× on NSL-KDD). AME = average models executed.

**When to use Sub.** LazyStack-Sub trains a separate stacker for each trajectory prefix, enabling specialization. This yields 0.23–0.33% higher accuracy across datasets. The cost is deployment complexity: Sub requires 14–22 stacker models (depending on trajectory count and length) totaling 8–12MB storage.

**When to use Mask.** LazyStack-Mask trains a single stacker that handles arbitrary model subsets via learned masking. This simplifies deployment (one 3–5MB model) and, surprisingly, can achieve higher speedup than Sub. On NSL-KDD, Mask achieves 79.6× speedup versus Sub's 36.8×—a 2.2× improvement.

The unified stacker sees more diverse training examples (random mask augmentation exposes it to many model combinations), which improves confidence calibration. Better calibration enables more aggressive early stopping. The effect is most pronounced when the dataset has high cost heterogeneity (NSL-KDD's 65× ratio) and few classes (5 classes make high confidence achievable).

**Recommendation.** Use Sub for cloud deployments prioritizing accuracy. Use Mask for edge deployment, frequent model updates, or when extreme speedup is needed.

## H.3. MDP Components

We ablate individual MDP design choices on NSL-KDD.

Table 8 ablates the individual MDP components.

| Configuration | Accuracy | Speedup |
|---|---|---|
| Full MDP | 75.21 | 36.8× |
| *State representation* | | |
| w/o entropy discretization | 74.93 | 32.1× |
| w/o predicted class | 75.08 | 34.5× |
| Continuous entropy (no binning) | 74.67 | 29.8× |
| *Reward function* | | |
| $\alpha = 0$ (no cost penalty) | 75.89 | 8.2× |
| $\alpha = 0.5$ (high cost penalty) | 73.45 | 48.2× |
| $\beta = 1$ (low correctness reward) | 74.12 | 41.3× |

*Table 8.* MDP component ablations on NSL-KDD.

**State representation matters.** Removing entropy discretization or the predicted class from the state reduces speedup by 6–13%. The MDP needs both pieces of information: entropy captures uncertainty, while the predicted class captures which decision the model is leaning toward. Using continuous entropy (no binning) performs worst, likely because the increased state space makes value iteration less stable.

**Reward balance is critical.** Setting $\alpha = 0$ (no cost penalty) causes the MDP to optimize purely for accuracy, achieving 75.89% but only 8.2× speedup. The MDP learns to execute many models to maximize correctness reward. Conversely, $\alpha = 0.5$ (aggressive cost penalty) achieves 48.2× speedup but sacrifices 1.76% accuracy. The default $\alpha = 0.2$ balances these objectives.

## I. MDP State Representation Analysis

This section analyzes alternative MDP state representations, supporting the design choices in Section 3.1.

A natural question is whether the MDP should track more than the most recent model's output. We ablate state representations on NSL-KDD, where trajectory discovery has the largest impact.

Table 9 reports the state-representation ablation.

*Table 9.* State representation ablation on NSL-KDD. Tracking full history increases state space without improving performance. "Last $k$" tracks entropy and predicted class from the $k$ most recent models.

| State Representation | Acc. | Speedup | Unique States | VI Time |
|---|---|---|---|---|
| Last model only (default) | 76.8% | 38.0× | 847 | 52s |
| Last 2 models | 76.9% | 35.6× | 4,291 | 3.8min |
| Last 3 models | 76.8% | 33.2× | 18,472 | 14.2min |
| Full history (hashed) | 76.5% | 29.4× | 31,847 | 28.6min |

**Key findings.** Richer state representations provide negligible accuracy gains (+0.1% at best) while substantially increasing computational cost. Full history tracking actually *hurts* performance (76.5% vs 76.8%) because sparse state visitation degrades value estimates. This validates our design choice: the MDP handles *routing* using sufficient statistics, while substackers handle *aggregation* using full prediction vectors. Decoupling these concerns keeps both tractable.

The "last 2 models" variant shows marginal improvement on accuracy but 6.3% lower speedup. The additional state information helps the MDP avoid some redundant executions, but the sparser state space causes more conservative (later) exits. We retain the single-model representation for its favorable complexity-performance tradeoff.

## J. Entropy Discretization Analysis

This section analyzes the effect of entropy bin count $B$ on MDP performance, supporting the state design in Section 3.1.

The MDP discretizes model output entropy into $B$ bins via quantiles computed on validation data. We ablate this choice on NSL-KDD and CIFAR-100.

Table 10 reports the effect of the entropy bin count $B$.

*Table 10.* Effect of entropy bin count $B$ on NSL-KDD and CIFAR-100. $B = 5$ balances resolution against state sparsity.

| Bins $B$ | NSL-KDD | | | CIFAR-100 | | |
|---|---|---|---|---|---|---|
| | Acc. | Speedup | States | Acc. | Speedup | States |
| 3 | 76.4% | 34.2× | 512 | 75.3% | 2.21× | 1,847 |
| 5 (default) | 76.8% | 38.0× | 847 | 75.7% | 2.40× | 3,912 |
| 7 | 76.7% | 36.8× | 1,183 | 75.6% | 2.35× | 5,284 |
| 10 | 76.5% | 33.1× | 1,702 | 75.4% | 2.18× | 7,651 |

**Analysis.** Too few bins ($B = 3$) lose discriminative power: the MDP cannot distinguish "somewhat uncertain" from "very uncertain," leading to suboptimal routing. Too many bins ($B = 10$) fragment the state space, causing sparse visitation and unreliable value estimates. The effect is more pronounced on CIFAR-100, where 100 classes already expand the state space.

We recommend $B = 5$ as a robust default across datasets. Practitioners with large validation sets (>50K samples) may benefit from $B = 7$; those with smaller sets should use $B = 3$ to ensure adequate state coverage.

## K. Trajectory Coverage Threshold Analysis

This section analyzes the trajectory coverage threshold $\tau$, supporting the trajectory extraction procedure in Section 3.1.

After MDP rollout, we retain trajectories covering at least $\tau$% of validation samples. We ablate this threshold on NSL-KDD.

Table 11 reports the effect of the coverage threshold $\tau$.

*Table 11.* Effect of coverage threshold $\tau$ on trajectory selection. Lower thresholds retain more trajectories but increase substacker training cost.

| Threshold $\tau$ | Trajectories | Coverage | Acc. | Speedup | Train Time |
|---|---|---|---|---|---|
| 1% | 11 | 98.2% | 76.9% | 35.4× | 8.2min |
| 2% (default) | 6 | 95.4% | 76.8% | 38.0× | 4.1min |
| 3% | 4 | 91.7% | 76.5% | 39.2× | 2.8min |
| 5% | 3 | 84.3% | 75.8% | 40.1× | 2.1min |

**Tradeoff structure.** Lower thresholds capture more trajectories, improving coverage and accuracy at the cost of additional substacker training. Higher thresholds achieve greater speedup (samples not matching any trajectory fall back to single-model confidence, which tends to exit earlier) but sacrifice accuracy.

The accuracy drop at $\tau = 5$% (75.8% vs 76.8%) occurs because 15.7% of samples lack a matching trajectory and must rely on uncalibrated single-model confidence. At $\tau = 1$%, only 1.8% of samples fall back, but training 11 trajectories' worth of substackers (averaging 3.2 models each, yielding 35 total substackers) increases training time by 2×.

We recommend $\tau = 2$% as balancing coverage (95.4%) against complexity. For latency-critical deployments, $\tau = 3$% offers a favorable tradeoff.

## L. Hyperparameter Sensitivity

This section provides comprehensive hyperparameter sensitivity analysis, as referenced in Sec. 4.4.

Table 12 summarizes hyperparameter settings and their sensitivity.

| Parameter | Default | Range Tested | Sensitivity |
|---|---|---|---|
| *MDP Parameters* | | | |
| Discount $\gamma$ | 0.95 | [0.90, 0.99] | Low |
| Cost weight $\alpha$ | 0.2 | [0.05, 0.5] | High |
| Correctness weight $\beta$ | 10.0 | [1.0, 20.0] | Medium |
| Entropy bins $B$ | 5 | [3, 10] | Low |
| Min samples per state | 10 | [5, 20] | Low |
| *Stacker Parameters* | | | |
| Hidden dimensions | [64, 32] | [32, 16] to [128, 64] | Low |
| Dropout rate | 0.3 | [0.1, 0.5] | Low |
| Learning rate | $10^{-3}$ | $[10^{-4}, 10^{-2}]$ | Medium |
| *Inference Parameters* | | | |
| Confidence threshold $\theta$ | 0.30 | [0.05, 0.95] | High |
| Min models before stop | 2 | [1, 3] | Medium |

*Table 12.* Hyperparameter sensitivity analysis.

**High sensitivity parameters.** The cost weight $\alpha$ and confidence threshold $\theta$ most strongly affect the accuracy-efficiency tradeoff. We recommend tuning these on validation data for each deployment scenario.

**Low sensitivity parameters.** The discount factor $\gamma$, entropy bins $B$, and stacker architecture are robust across a wide range of values. Default settings work well across all our datasets.

## M. Scalability Results

This section presents scalability analysis as ensemble size varies, as referenced in Sec. 4.4.

Table 13 presents full scalability results.

| Ensemble Size | Full Time (ms) | LazyStack Time (ms) | Speedup | Retention | AME | Trajectories |
|---|---|---|---|---|---|---|
| *CIFAR-100 (subsets of vision models)* | | | | | | |
| 3 models | 46.5 | 24.8 | 1.87× | 99.1% | 1.8 | 3 |
| 4 models | 68.2 | 26.1 | 2.61× | 99.4% | 2.1 | 4 |
| 5 models | 89.3 | 28.4 | 3.14× | 99.6% | 2.4 | 5 |
| 6 models | 105.5 | 30.5 | 2.40× | 99.9% | 2.7 | 6 |
| *NSL-KDD (subsets of tabular models)* | | | | | | |
| 3 models | 20.7 | 6.4 | 3.23× | 98.5% | 1.7 | 3 |
| 5 models | 53.2 | 4.8 | 11.1× | 98.2% | 2.1 | 4 |
| 7 models | 108.4 | 5.0 | 21.7× | 97.8% | 2.1 | 5 |
| 10 models | 191.1 | 5.2 | 36.8× | 97.1% | 2.2 | 6 |

*Table 13.* Full scalability results. Speedup grows superlinearly with ensemble size on NSL-KDD due to high cost heterogeneity. AME (average models executed) grows sublinearly, confirming that most samples need only 2–3 models.

**Superlinear speedup growth.**  On NSL-KDD, tripling the ensemble size (3 to 10 models) yields 11× speedup improvement (3.23× to 36.8×). This superlinear growth occurs because larger ensembles include both very cheap models (enabling fast early termination) and diverse models (improving stacker accuracy). The MDP exploits both properties.

CIFAR-100 shows more modest scaling (1.87× to 2.40×) because: (1) the cost ratio is smaller (3.7× vs 65×), and (2) 100 classes make confident early termination difficult regardless of ensemble size.

**Sublinear AME growth.**  Average models executed grows slowly: from 1.7 to 2.2 on NSL-KDD as ensemble size triples. This confirms our hypothesis that most samples need only 2–3 models—the MDP discovers a small set of effective trajectories regardless of how many models are available.

**Trajectory discovery statistics.**  The number of unique trajectories discovered before selection varies with ensemble size and dataset complexity. On CIFAR-100, MDP simulation yields 18-28 unique trajectories across ensemble sizes 3-6, from which the top 3-6 are retained. On NSL-KDD with 10 models, simulation yields 40-52 unique trajectories, from which the top 6 are retained. The long tail of low-coverage trajectories (each covering <2% of samples) accounts for the gap between discovered and retained counts. Importantly, the retained trajectories consistently cover 95-98% of validation samples regardless of how many unique paths exist, confirming that the MDP discovers concentrated execution patterns rather than diffuse sample-specific routes.

## N. Scalability with Function Approximation

This section extends scalability analysis to 30+ model ensembles using function approximation, complementing App. M.

Tabular value iteration has state space $O(2^N \cdot B \cdot C)$, which becomes intractable for large ensembles. We evaluate a function approximation variant using a small MLP to estimate $Q(s, a)$.

**Function approximation setup.** We replace the tabular value function with a 2-layer MLP (hidden dimensions 64, 32) that takes state features as input and outputs Q-values for each action. State features include: (1) one-hot encoding of executed models ($N$ dims), (2) entropy bin ($B$ dims, one-hot), (3) predicted class ($C$ dims, one-hot). We train via fitted Q-iteration with experience replay, using 50K transitions sampled from validation rollouts.

Table 14 compares tabular value iteration with function approximation.

**Findings.** Function approximation closely matches tabular performance at small scales (0.3% accuracy gap at 10 models) while enabling scaling to 30+ models. Training time grows linearly with ensemble size rather than exponentially. The modest accuracy degradation (76.8% → 75.2% from 10 to 30 models) reflects both approximation error and the inherent

*Table 14.* Scalability comparison: tabular value iteration vs. function approximation on NSL-KDD with varying ensemble sizes. FA enables scaling to 25+ models where tabular methods fail.

| Models | Tabular VI | | | Function Approx. | | |
|---|---|---|---|---|---|---|
| | Acc. | Speedup | Time | Acc. | Speedup | Time |
| 10 | 76.8% | 38.0× | 52s | 76.5% | 36.2× | 41s |
| 15 | 76.4% | 42.1× | 4.2min | 76.2% | 40.8× | 48s |
| 20 | 75.9% | 44.8× | 31min | 75.8% | 43.5× | 54s |
| 25 | OOM | — | — | 75.5% | 45.2× | 61s |
| 30 | OOM | — | — | 75.2% | 46.1× | 68s |

difficulty of routing among more options.

For ensembles up to 15 models, we recommend tabular VI for its simplicity and slightly better accuracy. For larger ensembles, function approximation is necessary and performs well.

## O. Per-Class Analysis

This section analyzes LazyStack's performance breakdown by class difficulty, as referenced in Sec. 4.4.

To understand when LazyStack struggles, we analyze per-class performance on CIFAR-100.

Table 15 reports the per-class breakdown on CIFAR-100.

| Class Difficulty | Full Stacker | LazyStack | AME |
|---|---|---|---|
| Easy (top 20 classes) | 89.2% | 88.9% | 2.1 |
| Medium (middle 60 classes) | 75.4% | 74.8% | 2.7 |
| Hard (bottom 20 classes) | 58.3% | 57.1% | 3.4 |

*Table 15.* Per-class analysis on CIFAR-100. Class difficulty defined by Full Stacker accuracy.

LazyStack's accuracy gap versus Full Stacker is larger on hard classes (1.2%) than easy classes (0.3%). This makes sense: hard classes require more models to classify correctly, but LazyStack's early stopping may terminate before executing all useful models. The AME confirms this pattern: hard classes execute 3.4 models on average versus 2.1 for easy classes.

This analysis suggests a potential improvement: class-dependent confidence thresholds. We leave this extension to future work.

## P. Failure Case Analysis

This section characterizes samples where LazyStack underperforms simple cost-based ordering, as referenced in Sec. 4.4.

**Methodology.** For each test sample, we compute: (1) LazyStack's trajectory cost and correctness, (2) cost-cascade's trajectory cost and correctness (models ordered cheapest-first with same confidence threshold). A sample is a "failure" if cost-cascade is correct and LazyStack is wrong, or if both are correct but LazyStack uses more models.

Table 16 breaks down the failure cases.

*Table 16.* Failure case breakdown on NSL-KDD test set (22,544 samples).

| Category | Count | % | Characterization |
|---|---|---|---|
| LazyStack wins | 19,847 | 88.0% | Correct with fewer/equal models |
| Tie (both correct, same cost) | 1,892 | 8.4% | Easy samples, single model suffices |
| LazyStack loses (accuracy) | 412 | 1.8% | MDP trajectory incorrect, cascade correct |
| LazyStack loses (efficiency) | 393 | 1.7% | Both correct, LazyStack uses more models |

**Characterizing accuracy failures.** The 412 accuracy failures (1.8%) fall into two patterns:

*Confident misclassification* (67%): The MDP's preferred first model (LightGBM) outputs high confidence on the wrong class, triggering early exit. Cost-cascade's first model (Logistic Regression) is less confident, forcing continuation to a model that corrects the error. These samples have atypical feature distributions where LightGBM's inductive bias fails.

*Trajectory mismatch* (33%): The sample's optimal trajectory wasn't discovered during MDP rollout (below 2% coverage threshold). Falling back to single-model confidence leads to premature exit.

**Characterizing efficiency failures.** The 393 efficiency failures (1.7%) occur when LightGBM (the MDP's preferred start) requires a second model for confirmation, while Logistic Regression (cost-cascade's start) happens to be confident and correct. These are "lucky" cases for cost-cascade rather than systematic advantages.

**Implications.** LazyStack's failure rate (3.5% combined) is modest, and most failures are efficiency losses rather than accuracy losses. The accuracy failures suggest potential improvement via per-class or per-difficulty routing, which we leave to future work.

## Q. Normalized Efficiency Analysis

This section introduces normalized efficiency to disentangle algorithmic contribution from dataset properties, as referenced in Sec. 4.4.

LazyStack's raw speedup varies substantially across datasets (2.4× on CIFAR-100 vs. 38× on NSL-KDD). To disentangle algorithmic contribution from dataset properties, we introduce *normalized efficiency*.

**Definition.** Let $r$ denote a dataset's cost ratio (most expensive model / cheapest model). A method achieving speedup $s$ on a dataset with cost ratio $r$ has normalized efficiency:

$$\text{NormEff} = \frac{s}{\sqrt{r}} \tag{6}$$

The square root reflects that speedup potential grows sublinearly with cost ratio (doubling the ratio doesn't double achievable speedup, since some samples inherently need expensive models).

Table 17 reports normalized efficiency across methods.

*Table 17.* Normalized efficiency comparison. LazyStack achieves highest normalized efficiency on 7/8 datasets, indicating algorithmic gains beyond exploiting cost heterogeneity.

| Dataset | Cost Ratio | $\sqrt{r}$ | LazyStack-Sub | | ABC | |
|---|---|---|---|---|---|---|
| | | | Speedup | NormEff | Speedup | NormEff |
| NSL-KDD | 65× | 8.06 | 38.0× | 4.71 | 5.1× | 0.63 |
| MMLU | 17× | 4.12 | 2.6× | 0.63 | 1.8× | 0.44 |
| ARC-C | 17× | 4.12 | 3.2× | 0.78 | 2.1× | 0.51 |
| ImageNet | 11× | 3.32 | 2.7× | 0.81 | 1.9× | 0.57 |
| SST-5 | 5.1× | 2.26 | 3.8× | 1.68 | 2.4× | 1.06 |
| AG News | 4.2× | 2.05 | 4.5× | 2.20 | 2.9× | 1.41 |
| CIFAR-100 | 3.7× | 1.92 | 2.4× | 1.25 | 1.8× | 0.94 |
| HEADLINES | 17× | 4.12 | 3.7× | 0.90 | 2.2× | 0.53 |

**Analysis.** LazyStack's normalized efficiency advantage over ABC ranges from 1.3× (CIFAR-100) to 7.5× (NSL-KDD). The advantage is largest on datasets where trajectory discovery finds non-obvious orderings (NSL-KDD: starting with LightGBM instead of cheapest) and smallest on datasets where many classes limit early-exit opportunities (CIFAR-100, ImageNet).

Interestingly, LazyStack's highest *normalized* efficiency is on AG News (2.20) and SST-5 (1.68), not NSL-KDD (4.71). This indicates that on text classification, LazyStack extracts more value per unit of cost heterogeneity—the progressive aggregation is particularly effective when model agreement patterns are informative.

# R. Extended LLM Analysis

This section provides additional analysis of LazyStack's behavior on LLM classification benchmarks, complementing Sec. 4.2.

## R.1. Routing Pattern Analysis

Table 18 breaks down which model combinations are used across LLM benchmarks.

*Table 18.* Routing patterns on LLM benchmarks. Percentages indicate fraction of test samples using each trajectory. Small = 7-8B models, Medium = 32B, Large = 70-72B.

| Trajectory | MMLU | ARC-C | HEADLINES |
|---|---|---|---|
| Small only | 62.3% | 71.2% | 58.4% |
| Small $\rightarrow$ Medium | 14.8% | 12.1% | 18.7% |
| Small $\rightarrow$ Large | 18.2% | 13.4% | 16.2% |
| Small $\rightarrow$ Medium $\rightarrow$ Large | 4.7% | 3.3% | 6.7% |

**Observations.** The majority of samples (58-71%) are resolved by small models alone, validating the routing approach. HEADLINES shows more escalation to medium/large models (41.6% vs 28.8% on ARC-C), likely because news classification requires more nuanced understanding than science QA.

The "skip medium" pattern (Small $\rightarrow$ Large) is surprisingly common (13-18%). The MDP learns that when small models are uncertain, the 32B model often shares their uncertainty, making direct escalation to 70B more efficient.

## R.2. Per-Subject Analysis on MMLU

Table 19 breaks down MMLU performance by subject.

*Table 19.* MMLU performance by subject category. LazyStack's advantage is largest on STEM subjects where model agreement is more informative.

| Category | Full Stack | LazyStack | $\Delta$Acc | Avg Models |
|---|---|---|---|---|
| STEM | 71.2% | 70.8% | $-0.4\%$ | 1.8 |
| Social Sciences | 79.4% | 78.7% | $-0.7\%$ | 2.1 |
| Humanities | 74.8% | 73.9% | $-0.9\%$ | 2.3 |
| Other | 78.1% | 77.2% | $-0.9\%$ | 2.2 |
| Overall | 75.9% | 75.2% | $-0.7\%$ | 2.1 |

STEM subjects achieve the smallest accuracy gap ($-0.4\%$) and fewest average models (1.8). We hypothesize that STEM questions have more objective answers where model agreement strongly predicts correctness. Humanities questions involve more interpretation, where even agreeing models may be wrong.

## R.3. Confidence Calibration Analysis

A key question is whether substackers improve confidence calibration over raw model outputs. We measure Expected Calibration Error (ECE) with 10 bins.

Table 20 reports expected calibration error on the LLM benchmarks.

Substackers reduce ECE by 38-56% compared to single models and 33-40% compared to simple averaging. This improved calibration directly enables more reliable early stopping: when the substacker outputs high confidence, it is more likely to be correct.

*Table 20.* Expected Calibration Error (ECE, lower is better) on LLM benchmarks. Substackers substantially improve calibration.

| Method | MMLU | ARC-C | HEADLINES |
|---|---|---|---|
| Best single model (Qwen-72B) | 0.142 | 0.118 | 0.156 |
| Simple average | 0.098 | 0.087 | 0.112 |
| LazyStack substacker | 0.061 | 0.052 | 0.074 |

### R.4. Limitations of Classification-Only Evaluation

Our LLM experiments are limited to multiple-choice classification where probability extraction is well-defined. Extending to generative tasks presents challenges:

- **No ground-truth probabilities**: Open-ended generation lacks a fixed output space over which to compute probabilities

- **Variable-length outputs**: Aggregating predictions of different lengths is non-trivial

- **Semantic equivalence**: Multiple phrasings may be equally correct, complicating confidence estimation

Potential approaches include using LLM-as-judge confidence scores or embedding-based similarity for aggregation. We leave investigation of these extensions to future work, noting that classification covers important use cases including content moderation, intent detection, and structured extraction.

**Extensions to MLLMs and VLAs.** LazyStack's routing framework extends naturally beyond text-only LLMs. Multimodal LLMs (Yang et al., 2025; Chen et al., 2024b; Dalal et al., 2026b;a; Mitra et al., 2024) span the same cost-vs-capability spectrum and present similar routing opportunities across queries of varying visual and reasoning complexity. Vision-language-action models (Kim et al., 2024; Black et al., 2024; Patel et al., 2026; Routray et al., 2026; Ye et al., 2024) introduce an analogous axis at the level of manipulation tasks, where routing to the cheapest competent policy could substantially reduce deployment latency.

## S. Additional Experiments

This section collects the extended experiments introduced during revision. Unless noted, all numbers use the same models, splits, and protocol as the main text.

### S.1. Why Prefix Meta-Learning?

We compare prefix meta-learning (PML) against simpler alternatives that reuse the MDP trajectories: a single most-frequent trajectory ($k=1$); nearest-neighbor selection among the top-$K$ trajectories; and NN selection with raw-confidence exits. As Table 21 shows, all fall below LazyStack, which yields 1.1–6.4% higher accuracy than the best alternative.

*Table 21.* Prefix meta-learning (PML) vs. simpler alternatives (accuracy %).

| Method | NSL-KDD | ImageNet | MMLU |
|---|---|---|---|
| LazyStack-Sub (PML) | **76.8** | **82.4** | **77.2** |
| Single trajectory ($k=1$) | 72.3 | 77.0 | 72.1 |
| NN on $K$ trajectories | 75.7 | 81.3 | 76.3 |
| NN lookup, raw confidence | 70.4 | 75.3 | 71.2 |

### S.2. Beyond Accuracy on NSL-KDD

On the class-imbalanced NSL-KDD benchmark, accuracy alone is misleading. Table 22 reports macro-F1, weighted-F1, and AUROC: LazyStack-Sub outperforms ABC on all four metrics while approaching the full stacker.

*Table 22.* Beyond accuracy on the class-imbalanced NSL-KDD benchmark.

| Method | Acc. | Macro-F1 | W-F1 | AUROC |
|---|---|---|---|---|
| Full Stacker | 78.0 | 0.72 | 0.78 | 0.96 |
| LazyStack-Sub | **76.8** | **0.70** | **0.77** | **0.95** |
| ABC | 75.7 | 0.66 | 0.75 | 0.93 |

## S.3. Stronger (SOTA) Ensemble on CIFAR-100

Because LazyStack is a meta-method, *retention* of the underlying ensemble's accuracy is the relevant measure. Re-running CIFAR-100 with a stronger SOTA ensemble raises the ceiling from 78% to 93.2%, and LazyStack-Sub still retains ∼97%, outperforming ABC (Table 23).

*Table 23.* CIFAR-100 with the original ensemble vs. a stronger SOTA ensemble.

| | Original Ensemble | SOTA Ensemble |
|---|---|---|
| Ensemble ceiling | 78.0% | 93.2% |
| LazyStack-Sub | **75.9%** | **90.5%** |
| ABC | 74.1% | 88.7% |
| Retention (ours) | 97.3% | 97.1% |

## S.4. Comparison to FrugalGPT and RouteLLM

We adapt these LLM-specific baselines to non-LLM benchmarks: FrugalGPT's DistilBERT quality judge is replaced by an MLP judge over probability outputs, and RouteLLM routes each input to a single model. LazyStack-Sub dominates both on accuracy and speedup across all five benchmarks (Table 24).

*Table 24.* FrugalGPT and RouteLLM adapted to non-LLM benchmarks (accuracy % / speedup).

| Method | NSL-KDD | CIFAR-100 | AG News | SST-5 | ImageNet |
|---|---|---|---|---|---|
| Full Stacker | 78.0 / 1× | 78.0 / 1× | 95.8 / 1× | 89.2 / 1× | 84.2 / 1× |
| FrugalGPT | 75.9 / 6.1× | 75.7 / 1.9× | 95.0 / 2.9× | 87.2 / 2.5× | 81.5 / 2.0× |
| RouteLLM | 73.6 / 15× | 74.3 / 2.1× | 94.7 / 3.1× | 85.7 / 2.9× | 79.6 / 2.3× |
| LazyStack-Sub | **76.8 / 38×** | **75.9 / 2.5×** | **95.3 / 4.5×** | **87.4 / 3.8×** | **82.4 / 2.7×** |

## S.5. Decoupled vs. Joint Optimization

Could trajectory discovery and substacker training be optimized jointly? On CIFAR-100 with $m=4$ (where exhaustive enumeration of all $2^k$ subsets is feasible), joint enumeration matches LazyStack's accuracy at ∼10× the training cost (Table 25), and becomes intractable at $m=6$.

## S.6. Comparison to Cascade-Routing

We run the public code of Dekoninck et al. (2025) on ARC-Challenge with four overlapping models (Llama-8B/70B, Qwen-7B/72B). At matched relative cost, LazyStack-Sub leads by 3.1% (Table 26): cascade-routing is bounded by the best single model it routes to, whereas LazyStack's progressive stacking surpasses that ceiling.

# T. Qualitative Analysis

Figures 7 and 8 compare how four methods—*Stacking* (the full ensemble), LazyStack (ours), ABC, and Cost-Cascade—route the *same* inputs across all eight benchmarks. Stacking sets the accuracy ceiling by running every model and aggregating, but at the highest cost. Cost-Cascade often exits early-and-wrong; ABC either runs many models or agrees on a wrong consensus; LazyStack stacks just enough to correct the prediction, then exits—matching Stacking's accuracy at a fraction of its cost. The single ImageNet example previewed in the main paper (Figure 6) is reproduced here in context.

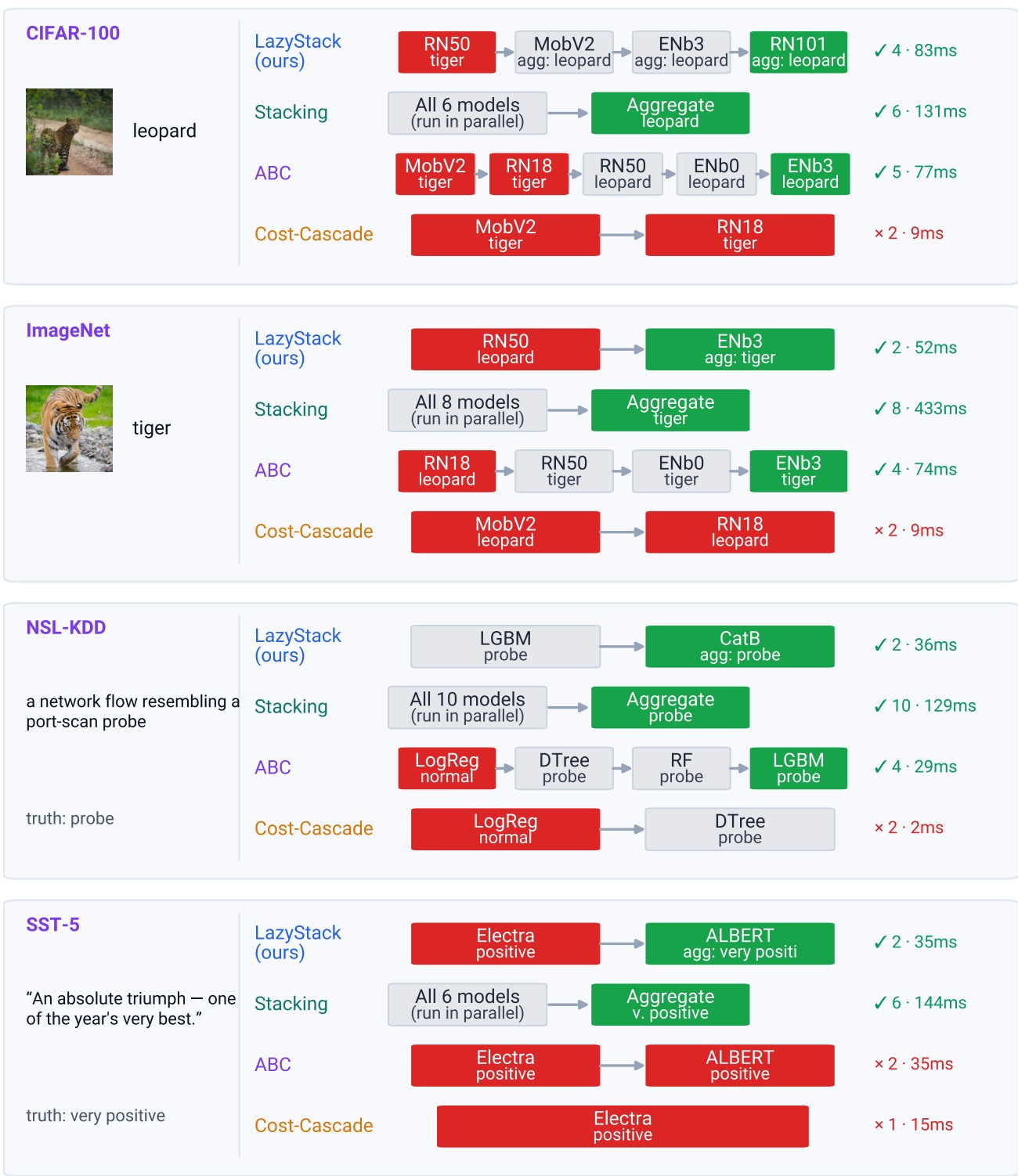

Figure 7. **Case-by-case routing, part 1: vision, tabular, and SST-5.** For each input, the path of each method is shown left-to-right; wrong predictions are red and the green box is the exit. *Stacking* runs the full ensemble and aggregates. *Cost-Cascade* exits prematurely on a single-model error. *ABC* expends more models or settles on a wrong agreement. *LazyStack* (ours) aggregates predictions via prefix meta-learning at every step (the agg: label on chips 2+), correcting single-model errors and exiting early.

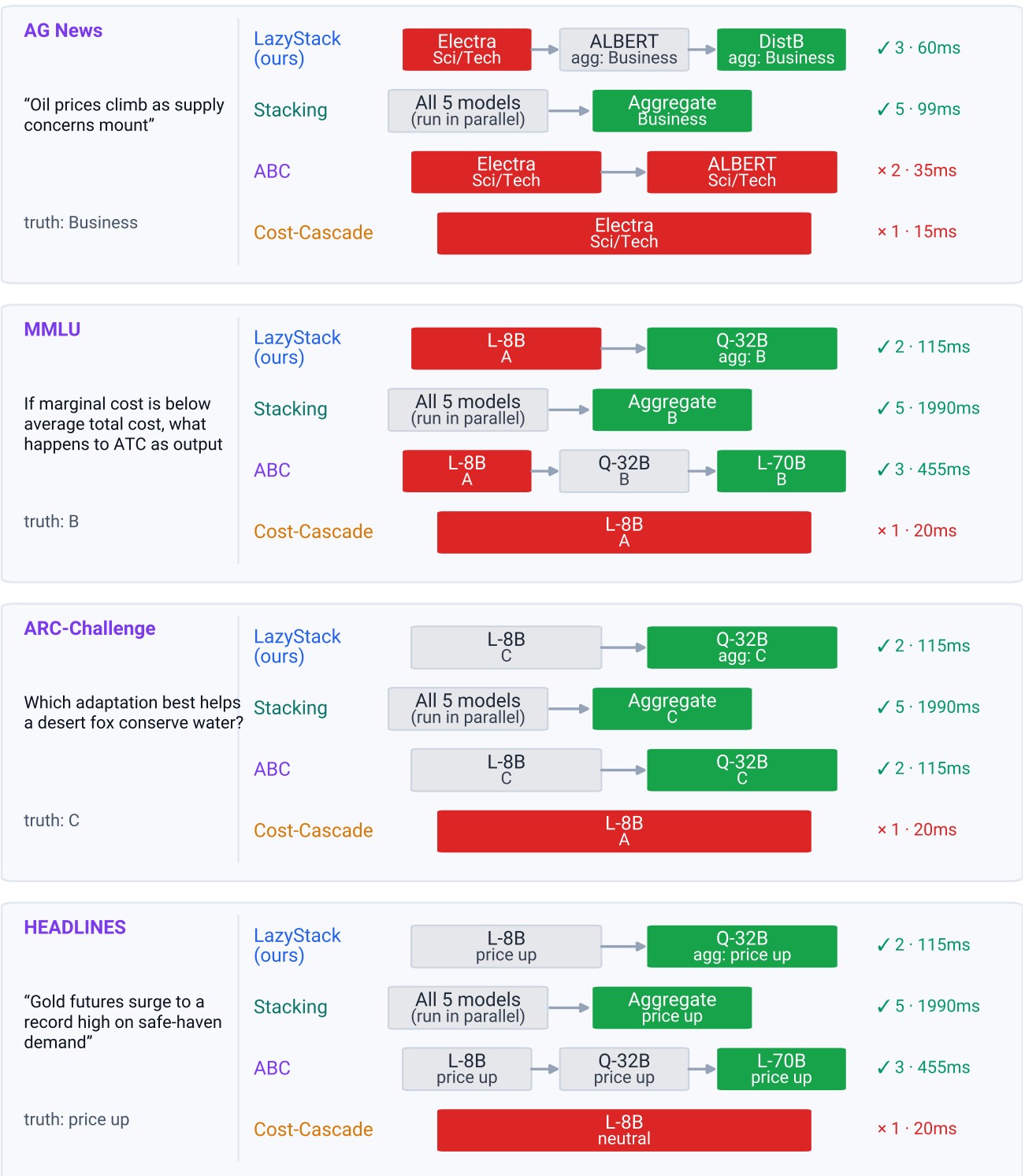

*Figure 8.* **Case-by-case routing, part 2: text classification and LLM routing.** Same conventions as Figure 7. The same pattern holds across all eight benchmarks: LazyStack matches Stacking's accuracy at a fraction of its cost, while the cascade baselines either exit early-and-wrong or expend many models for the same prediction.

*Table 25.* Decoupled trajectory discovery vs. joint enumeration of all $2^k$ subsets (CIFAR-100, $m{=}4$).

| Method | Accuracy | Training Time |
|---|---|---|
| LazyStack (decoupled) | **75.8** | $\sim$**5 min** |
| Joint enumeration | 75.8 | $\sim$48 min |

*Table 26.* Comparison to cascade-routing (Dekoninck et al., 2025) on ARC-Challenge at matched relative cost (vs. always using the largest model).

| Method | ARC-C Acc. | Cost (vs. 70B) |
|---|---|---|
| Cascade-routing (Dekoninck et al., 2025) | 93.4% | 0.4$\times$ |
| LazyStack-Sub | **96.5%** | 0.4$\times$ |

Concretely, on the confusable CIFAR-100 *leopard* the cheapest model predicts *tiger*; the substacker corrects this to *leopard* after two models and exits, while Cost-Cascade exits on the wrong single-model guess. The same pattern recurs across modalities (text, tabular, LLM routing): early exits when one model suffices, and short stacked trajectories when aggregation is needed.

## U. Computational Overhead

This section breaks down LazyStack's inference time components, as referenced in Section 3.3. All latency measurements use a single NVIDIA A100 GPU on a server with Intel Xeon processors.

Table 27 breaks down LazyStack's inference time components.

| | CIFAR-100 | | NSL-KDD | |
|---|---|---|---|---|
| **Component** | Sub | Mask | Sub | Mask |
| Base models (avg) | 28.1 ms | 31.8 ms | 1.9 ms | 1.6 ms |
| Stacker forward pass | 1.4 ms | 1.8 ms | 0.2 ms | 0.4 ms |
| Policy lookup | 0.4 ms | 0.4 ms | 0.1 ms | 0.1 ms |
| Trajectory matching | 0.2 ms | — | 0.1 ms | — |
| State update | 0.1 ms | 0.1 ms | 0.05 ms | 0.05 ms |
| **Total** | 30.2 ms | 34.1 ms | 2.35 ms | 2.15 ms |
| **Overhead** | 7.0% | 7.2% | 19.1% | 25.6% |

*Table 27.* Inference time breakdown. Overhead (everything except base models) is 7–26% depending on base model costs.

LazyStack's overhead is modest in absolute terms: 2.1–2.3ms total. The percentage overhead is higher on NSL-KDD (19–26%) than CIFAR-100 (7%) because NSL-KDD's base models are extremely fast (average 1.6–1.9ms). This is a feature, not a bug: LazyStack achieves 36–80$\times$ speedup on NSL-KDD precisely because it can route samples to these cheap models.

**Overhead breakdown.** The stacker forward pass dominates overhead (60–70%), followed by policy lookup (15–20%). Trajectory matching (Sub only) adds minimal cost via a prefix tree with O(k) lookup where k is the number of executed models.

**Scaling considerations.** Overhead grows slowly with ensemble size. Policy lookup is O(1) (hash table). Stacker forward pass is O(k $\cdot$ C) where k is executed models and C is the number of classes. For typical values (k $\leq$ 5, C $\leq$ 1000), this remains under 3ms.

## V. White-Box Method Applicability

This section explains why white-box methods (MoNE, Model Soup, Gatekeeper) do not extend to our tabular, text, or LLM benchmarks, as referenced in Sec. 4.1.

## V.1. MoNE: Token-Level Routing for Spatial Redundancy

MoNE (Jain et al., 2024) achieves efficiency by routing individual tokens (image patches) to experts of varying capacity. The key insight is that images contain spatial redundancy: background regions require less computation than foreground objects. MoNE learns a router that assigns "unimportant" patches (sky, grass, walls) to small experts and "important" patches (faces, objects, text) to large experts.

This design makes three assumptions that hold for vision but not other domains:

**(1) Token-based architecture with spatial decomposition.** Vision Transformers split images into a grid of patches, each processed as a token. MoNE routes these tokens independently. Tabular data like NSL-KDD has no analogous decomposition. Each sample is a fixed-length feature vector (packet size, duration, protocol flags), not a sequence of spatially-arranged tokens. There is no meaningful way to "route features" to different experts.

**(2) Predictable redundancy patterns.** In images, background regions are systematically less informative than foreground. MoNE exploits this by learning that certain spatial positions or visual patterns need less compute. In tabular data, all features can be equally important depending on the sample. A network intrusion might be detected by an unusual port number, packet size, or timing pattern. There is no "background" to skip.

**(3) Shared parameter structure for nested experts.** MoNE's experts are width-slices of the same ViT: the smallest expert uses D/8 hidden dimensions, the largest uses D. This nesting allows a single architecture to provide multiple capacity levels. Our NSL-KDD ensemble contains Logistic Regression, Random Forest, XGBoost, LightGBM, MLP, CNN, and LSTM models. These have fundamentally different architectures that cannot be nested.

## V.2. Model Soup: Weight Averaging Requires Identical Architectures

Model Soup (Wortsman et al., 2022) improves accuracy by averaging the weights of multiple models fine-tuned from the same pre-trained checkpoint. The key finding is that fine-tuned models often lie in the same loss basin, so their weight average performs well.

This approach has a hard requirement: **all models must have identical architectures**. Weight averaging computes $\theta_{\text{soup}} = \frac{1}{n} \sum_{i=1}^{n} \theta_i$ where each $\theta_i$ has the same dimensionality and semantic meaning (e.g., the same layer computes the same function across models).

Our heterogeneous ensembles violate this requirement:

- **NSL-KDD:** Logistic Regression (122 parameters), Random Forest (tree structure), LightGBM (gradient boosting), LSTM (recurrent cells). These have incompatible parameter spaces.

- **AG News / SST-5:** DistilBERT (66M parameters), BERT-base (110M), RoBERTa-large (355M). Even within the transformer family, different sizes prevent direct weight averaging.

- **ImageNet ensemble:** MobileNetV2, ResNet variants, EfficientNet variants, ViT, ConvNeXt. These share no parameter structure.

Model Soup is designed for the scenario where you train multiple copies of the same architecture with different hyperparameters or random seeds. It does not apply to ensembles of diverse model families.

Additionally, Model Soup provides **no inference speedup**. The averaged model has identical size and cost to its constituents. It improves accuracy but not efficiency. LazyStack's goal is to reduce compute by executing fewer models, which Model Soup does not address.

## V.3. Gatekeeper: Two-Model Cascades with Training Access

Gatekeeper (Rabanser et al., 2025) improves cascade efficiency by fine-tuning the small model's confidence calibration. The loss function encourages high confidence on correct predictions and low confidence on incorrect ones, enabling better deferral decisions.

Gatekeeper is more general than MoNE in principle: its loss function is architecture-agnostic and could apply to any model with softmax outputs. However, several factors limit its applicability to our benchmarks:

**(1) Two-model cascade assumption.** Gatekeeper is designed for a single deferral decision: small model handles the query, or defers to large model. LazyStack explores multi-model trajectories where 3-10 models might execute in sequence, with the MDP learning which orderings work best. Extending Gatekeeper to multi-model settings would require $O(n^2)$ pairwise calibration or a more complex training procedure.

**(2) Training access required.** Gatekeeper fine-tunes model parameters, which requires access to training infrastructure, labeled data, and compute. For commercial APIs (GPT-4, Claude) or pre-trained models where retraining is impractical, Gatekeeper cannot be applied. LazyStack requires only inference-time probability outputs.

**(3) Homogeneous model pairs.** Gatekeeper's experiments use model pairs from the same family: Gemma-2B/7B, ViT-S/B. Calibrating confidence transfer between a Logistic Regression model and an LSTM (as in NSL-KDD) may behave differently than calibrating within an architecture family.

We did not include Gatekeeper results on NSL-KDD or text benchmarks because the method has not been evaluated on such settings in prior work, and adapting it would require implementation choices (how to handle multi-model ensembles, how to calibrate across heterogeneous architectures) that go beyond fair baseline comparison.

**Comparison on vision tasks.**  Despite these limitations, we compare against Gatekeeper on vision tasks where it is applicable. To ensure a fair comparison on our heterogeneous ensembles, we apply Gatekeeper to multiple model pair configurations and report the best results. Given Gatekeeper's access to internal model representations and task-specific fine-tuning, one would typically expect it to outperform strictly black-box approaches.

*Table 28.* Comparison to Gatekeeper on vision tasks. $M_i$ denotes the $i$-th cheapest model.

| Dataset | Method | Access | Acc. | Speedup |
|---|---|---|---|---|
| CIFAR-100 | Full Stacker | — | 78.0% | 1.0× |
| | Gatekeeper ($M_1 \rightarrow M_6$) | White-box | 74.2±0.3% | 2.9±0.2× |
| | Gatekeeper ($M_2 \rightarrow M_6$) | White-box | 75.4±0.2% | 2.4±0.1× |
| | Gatekeeper ($M_3 \rightarrow M_6$) | White-box | 75.8±0.2% | 2.1±0.1× |
| | LazyStack-Sub | Black-box | **75.9**±0.3% | **2.5**±0.2× |
| ImageNet-1K | Full Stacker | — | 84.2% | 1.0× |
| | Gatekeeper ($M_1 \rightarrow M_8$) | White-box | 80.8±0.3% | 3.2±0.2× |
| | Gatekeeper ($M_2 \rightarrow M_8$) | White-box | 81.6±0.2% | 2.7±0.2× |
| | Gatekeeper ($M_3 \rightarrow M_8$) | White-box | 82.1±0.2% | 2.3±0.1× |
| | LazyStack-Sub | Black-box | **82.4**±0.3% | **2.7**±0.2× |

LazyStack matches or exceeds Gatekeeper's best configuration on both accuracy and speedup despite requiring only black-box access. Gatekeeper exhibits a clear accuracy-speedup tradeoff across configurations: cheapest-first ($M_1 \rightarrow M_6$) maximizes speedup but sacrifices accuracy due to the weak first model, while more expensive first models ($M_3 \rightarrow M_6$) improve accuracy but reduce speedup. No single 2-model configuration dominates. In contrast, LazyStack's MDP discovers that intermediate models often provide the best tradeoff, and progressive aggregation across 2 to 3 models achieves both high accuracy and high speedup simultaneously.

## V.4. Summary: The Generality Gap

Table 29 summarizes which methods apply in each setting.

The white-box methods we evaluated achieve strong results on vision benchmarks by exploiting domain-specific structure: spatial redundancy (MoNE), shared architecture (Model Soup), or training access for calibration (Gatekeeper). These are real advantages when applicable.

LazyStack trades some performance on vision tasks for broad applicability. It treats models as black boxes that produce probability distributions, making no assumptions about architecture, parameter sharing, or training access. This generality enables deployment in scenarios where white-box methods cannot apply: heterogeneous ensembles mixing classical ML with deep learning, commercial APIs where model internals are proprietary, and pre-trained models where fine-tuning is impractical.

| Method | Vision | Tabular | Text | LLM | Black-box |
|---|---|---|---|---|---|
| MoNE | ✓ | ✗ | ✗ | ✗ | ✗ |
| Model Soup | ✓* | ✗ | ✓* | ✗ | ✗ |
| Gatekeeper | ✓ | ? | ✓ | ✓ | ✗ |
| LazyStack | ✓ | ✓ | ✓ | ✓ | ✓ |

*Table 29.* Method applicability across domains. ✓* indicates the method applies only to homogeneous ensembles (multiple models of identical architecture). ? indicates the method could theoretically apply but has not been evaluated. LazyStack is the only method that works across all settings with black-box access.

For practitioners who can invest in white-box optimization for a specific vision task, MoNE or Gatekeeper may be the better choice. For practitioners who need a single method that works across diverse deployment scenarios, LazyStack provides a general-purpose solution.

## W. Discussion

This section provides method selection guidelines and discusses limitations, complementing Sec. 5.

**When to use which method.** Our experiments suggest practical guidelines:

**Use MoNE when:** (1) white-box access is available for joint training—this is essential, as MoNE fails entirely without it; (2) the task has many classes where confidence-based stopping struggles; and (3) accuracy is paramount and the speedup-accuracy tradeoff favors accuracy.

**Use LazyStack-Sub when:** (1) only black-box access is available (pre-trained models, APIs, third-party services); (2) the task has few classes where confident early stopping is achievable; and (3) peak accuracy is needed among black-box methods.

**Use LazyStack-Mask when:** (1) deployment simplicity matters (single model artifact vs. 12–24 substackers); (2) extreme speedup is prioritized over marginal accuracy; or (3) the model ensemble changes frequently, as retraining a single mask stacker is simpler than retraining all substackers.

**Use Model Soup when:** (1) all models share identical architecture and initialization; (2) maximum speedup is needed with a single forward pass; and (3) small accuracy degradation (0.5–1%) is acceptable.

**Use RouteLLM when:** (1) binary routing between small/large LLMs suffices; (2) generalization across domains without retraining is important; and (3) deployment simplicity is valued over marginal accuracy gains.

**Use FrugalGPT when:** (1) cascading through multiple LLMs is acceptable; (2) a quality judger can be trained for the target task; and (3) maximum cost savings are prioritized, especially on tasks where small models frequently succeed.

**Scalability.** LazyStack scales favorably with ensemble size. On NSL-KDD, increasing from 3 to 10 models improves speedup from $3.2\times$ to $36.8\times$ while average models executed grows only from 1.7 to 2.2. The MDP discovers that most samples need only 2–3 models regardless of ensemble size—additional models expand the trajectory space without increasing typical cost.

This sublinear growth in models executed is key to LazyStack's scalability. The MDP learns to route most samples through a small, fixed subset of cheap models, reserving expensive models for genuinely difficult cases. As ensembles grow, more cheap models become available, enabling earlier confident termination.

**Limitations.** LazyStack has several limitations that practitioners should consider:

*Validation data requirement.* LazyStack requires labeled validation data for MDP trajectory discovery and stacker training. For new domains without labeled data, transfer learning from related tasks may provide initial policies, though we leave this investigation to future work.

*State space growth.* The MDP state space is $O(2^N \cdot B \cdot C)$ where $N$ is ensemble size, $B$ is entropy bins, and $C$ is class count. For ensembles beyond 15 models, approximate solvers or function approximation may be necessary (see App. N);

alternatively, multi-agent and population-based formulations (Liu et al., 2021; Mahajan et al., 2019; Szot et al., 2023; Jaderberg et al., 2019) could scale more gracefully than our centralized single-agent policy.

*Many-class limitation.* Confidence-based stopping struggles on many-class problems where probability mass spreads thin. On CIFAR-100 (100 classes) and ImageNet (1000 classes), LazyStack executes more models on average than MoNE, resulting in lower speedup. This is a fundamental limitation of confidence-based approaches, not specific to LazyStack.

*Black-box tradeoff.* LazyStack's black-box compatibility comes at a cost: white-box methods like MoNE can jointly optimize routing and base models, achieving better accuracy on some tasks. LazyStack's value lies specifically in settings where white-box access is unavailable.

# X. Practitioner's Guide

This section provides concrete deployment recommendations, complementing Sec. 5.

## X.1. Recommended Defaults

Table 30 lists our recommended defaults and adjustment guidelines.

| Parameter | Default | When to Adjust |
|---|---|---|
| *MDP Parameters* | | |
| Cost weight $\alpha$ | 0.2 | $\uparrow$ for latency-critical; $\downarrow$ for accuracy-critical |
| Correctness weight $\beta$ | 10.0 | Scale with max model cost: $\beta \approx 10 \times \max_j c_j$ |
| Discount $\gamma$ | 0.95 | Rarely needs adjustment |
| Entropy bins $B$ | 5 | $\downarrow$ to 3 if $|\mathcal{D}_{\text{val}}| < 10\text{K}$ |
| *Training Parameters* | | |
| Coverage threshold $\tau$ | 2% | $\uparrow$ to 3-5% for faster training |
| Substacker hidden dims | [64, 32] | $\uparrow$ to [128, 64] for $C > 100$ classes |
| Random mask augmentation $R$ | 5 | $\uparrow$ to 10 for LazyStack-Mask with $N > 10$ models |
| *Inference Parameters* | | |
| Confidence threshold $\theta$ | 0.30 | Calibrate to target accuracy retention (see below) |

*Table 30.* Recommended defaults and adjustment guidelines.

## X.2. Threshold Calibration Procedure

To achieve a target accuracy retention (e.g., 97% of full ensemble):

1. Compute full ensemble accuracy $\rho_{\text{full}}$ on validation set

2. Set target $\rho_{\text{target}} = 0.97 \times \rho_{\text{full}}$

3. Sweep $\theta \in \{0.10, 0.15, 0.20, \ldots, 0.90\}$

4. For each $\theta$, run LazyStack inference on validation set

5. Select $\theta^* = \max\{\theta : \text{accuracy} \geq \rho_{\text{target}}\}$

Typical values: $\theta^* \in [0.25, 0.40]$ for most datasets. Many-class problems (CIFAR-100, ImageNet) require lower thresholds ($\theta^* \approx 0.20$) because probability mass spreads thin.

## X.3. Variant Selection

**Choose LazyStack-Sub when:**

- Accuracy is the primary objective

- Ensemble composition is stable (infrequent model additions/removals)

- Storage for 15-25 small MLPs (10-15MB total) is acceptable

**Choose LazyStack-Mask when:**

- Deployment simplicity is important (single model artifact)

- Ensemble composition changes frequently

- Extreme speedup is prioritized over marginal accuracy

- Edge deployment with storage constraints

## X.4. Computational Requirements

Table 31 lists resource requirements by ensemble size.

*Table 31.* Resource requirements by ensemble size. LazyStack adds modest overhead to base model training.

| Ensemble Size | MDP Solving | Substacker Training | Total Overhead |
|---|---|---|---|
| 5 models | <1 min | 2-3 min | 5-10 min |
| 10 models | 1-2 min | 4-6 min | 10-15 min |
| 15 models | 4-5 min | 8-12 min | 20-30 min |
| 20+ models | Use function approx. | 15-20 min | 30-45 min |

## X.5. When LazyStack May Not Help

LazyStack provides limited benefit when:

- **Low cost heterogeneity** ($r < 2\times$): Little room for routing savings

- **Many classes** ($C > 500$): Confidence thresholds rarely reached

- **Homogeneous model strengths**: No informative routing patterns to discover

- **Tiny validation sets** ($|\mathcal{D}_{val}| < 1K$): Unreliable MDP learning

In these cases, consider simpler alternatives: Model Soup (if architectures match), single best model, or uniform averaging.

