# OpenReview forum: "Unifying Stacking and Cascading for Efficient Ensemble Inference"
_ICML.cc/2026/Conference — ICML 2026 regular_

### Official Review · Reviewer_STtx · 2026-02-25

**Soundness:** 2
**Presentation:** 2
**Significance:** 3
**Originality:** 3
**Overall Recommendation:** 4
**Confidence:** 1

**Summary:**

The paper addresses the issues of the traditional ensembles. Stacking runs all models and aggregates predictions, leading to high accuracy but high cost. Cascades run models sequentially and exit early, leading to lower cost but inaccurate accuracy. The motivation of the paper is to achieve ensemble-level accuracy without running all models on every input. It proposes LazyStack that computes confidence of early exit. The key is to use MDP to formulate the trajectory discovery of the sequence of the executing the models and then train a substackers to output ensemble prediction and exit confidence.

**Compliance With Llm Reviewing Policy:**

Affirmed.

**Key Questions For Authors:**

I may have missed the answers for the following questions, would you like to point me out or answer the questions:

How are the executed models represented in the MDP? Are the trained MDP and substackers learned to fit the given set of the models and the test data? If so, isn’t that enough to get the ensemble results as the models execute when training MDP and substackers? If not, are the MDP and substackers trained?

What are the H(p)? Why do you select h and k as the state?

What does it mean that “95%+ of the samples across all 8 datasets follow just 3-8 trajectories?” My understanding is that each model evaluates on all or at least some of the data in the dataset and then move on to the next model?

**Limitations:**

Yes.

**Strengths And Weaknesses:**

**Strength**
The paper is well-motivated and the background issues of the previous methods are well articulated. The proposed method in general is clearly described. The advantages of using the method are supported by the experiments.

**Weakness**
Some details of the method are not clear to me. See Questions.

---

> ### Author Rebuttal · Authors · 2026-03-31
>
> We have addressed all 3 reviewer concerns by clarifying our design choices and referencing specific sections of the manuscript. Since several questions pertain to the MDP setup, we provide a concrete walkthrough below to illustrate the execution flow.
>
> ​​Example:  Consider an ensemble of three models: Logistic Regression (0.8ms), LightGBM (2.1ms), and SVM (18.3ms). Executing the full ensemble requires approx.. 21.2ms (0.8 + 2.1 + 18.3), assuming negligible cost for intermediate stackers.
>
> 1. Upon receiving a new test input, the MDP policy selects a model—for instance, LightGBM. This may not be the cheapest option; the MDP learns an optimized ordering based on the data (§4.4, L373)
> 2. Say LightGBM outputs class probabilities . [0.85, 0.05, 0.04, 0.03, 0.03]. The state becomes s = (E={LightGBM}, h=low entropy, k=class 1). We check its raw confidence: 0.85 > θ (where θ=0.30). We, thus,  exit immediately with the latency of 2.1ms instead of 21.2ms.
> 3. For a harder input (e.g., confidence 0.22) which is less than θ=0.30, the policy triggers the next model. If a substacker exists for the resulting model  prefix P   - i.e.,   P belongs to the set of prefixes, identified during trajectory discovery,  that cover 95%+ of samples, §3.2, L251)  --  the system aggregates both predictions, and outputs calibrated confidence.
> 4. If the calibrated confidence > θ, we exit; otherwise we continue as detailed in Algorithm 1, L275.
>
> **[Q1]** *How are executed models represented in the MDP? Are MDP and substackers fit to test data?*
>
> In the state s = (E, h, k), E is the set of executed model indices. In the example above, E={LightGBM} after the first step. This tells the policy which models have run; h and k summarize what the most recent execution revealed (see Q2 below). The MDP policy is solved via value iteration on the state space defined by these tuples (§3.1, L203).
>
> To directly address the concern: MDP and substackers are trained exclusively on train/validation splits, never on test data. For all benchmarks, we use standard train/test splits (App B, L632; App D, L745):
>
> | Component | Data Source | Reference |
> |---|---|---|
> | MDP transitions (Eq. 1) | Validation split | §3.1, L216 |
> | MDP policy (Eq. 3) | Value iteration on above | §3.1, L234 |
> | Substackers | Validation split | §3.2, L251 |
> | Threshold calibration | Validation split | §3.3, L298 |
> | Evaluation | Test split (never seen in training) | §4, L317 |
>
> As with all adaptive inference methods in this family (FrugalGPT 2024; RouteLLM 2025; Gatekeeper 2025), everything is learned offline. The efficiency gain comes at inference on unseen inputs, where LazyStack routes each sample through only 2 to 3 models on average via the learned policy and substackers (Algorithm 1, L275; §4.3, L356).
>
> **[Q2]** *What is H(p)? Why select h and k as the state?*
>
> H(p) is the standard Shannon entropy of a model's output probability distribution: H(p) = −Σ pᵢ log pᵢ. It measures prediction uncertainty: a confident output like [0.85, 0.05, ...] has low entropy, while a spread-out output like [0.25, 0.20, ...] has high entropy. We discretize H(p) into B=5 bins (§3.1, L212).
>
> We select h and k because they capture the two signals needed for routing: h tells the policy *how uncertain* the current prediction is (should we continue?), and k tells it *which class* is predicted (some classes are systematically harder and benefit from additional models). In the walkthrough above, low h triggers early exit; high h triggers continuation.
>
> We chose this minimal representation over richer alternatives deliberately. Appendix I (Tab 9) ablates tracking more history (last 2 to 3 models): the state space grows 5 to 20x with no benefit to accuracy or speedup, because substackers already handle full prediction aggregation (§3.2, Fig 2b). The MDP only needs to decide *routing*; aggregation is handled separately.
>
> **[Q3a]** “Sequential vs Parallel Model Execution?”
>
> To clarify: models execute sequentially on each individual input, exiting as soon as confidence is reached (please see the example).
>
> **[Q3b]** *What does "95%+ of samples follow 3-8 trajectories" mean?*
>
> This is referring to the observation that when we execute models based on (learned) MDP policy on the validation dataset,  the  sequence of models that execute prior to exit (i.e, resulting trajectories) are only a few  (3-8 depending on the benchmark). To further clarify, in LazyStack:
>
> 1. We first learn the MDP policy via value iteration on validation data (§3.1, L203).
> 2. We then roll out this policy on the validation set. Each input produces a trajectory, e.g., [LightGBM, CatBoost, exit], depending on when confidence is reached.
> 3. We rank trajectories by how many inputs follow them, and keep the top-K covering 95%+ of the data. This turns out to be just 3 to 8 trajectories (App K)
> 4. We train substackers for every prefix of these top-K trajectories (§3.2, L251). This reduces  # substacker learned from 2^k to linear per trajectory.

---

> > ### Author Rebuttal · Reviewer_STtx · 2026-03-31
> >
> > Most of my concerns are writing and logical reasoning how the decisions are made. The rebuttal clarify my concerns. Please incorporate the clarification to the draft and thank you for your explanations.

---

> > > ### Author Response · Authors · 2026-04-01
> > >
> > > Thank you for your thoughtful review and engagement with our work. If there’s anything we can clarify or provide to help address any remaining concerns or improve your assessment, we’d be glad to do so. We truly appreciate your time and feedback.

---

### Official Review · Reviewer_3vfD · 2026-03-07

**Soundness:** 3
**Presentation:** 3
**Significance:** 3
**Originality:** 2
**Overall Recommendation:** 5
**Confidence:** 4

**Summary:**

This work targets an interesting problem of unifying model cascading with ensembling. The proposed LazyStack method learns ensemble aggregators with an efficient trajectory concentration approach, and uses value iteration to optimize model orderings in the cascade. The experiments on several datasets show good accuracy-speedup tradeoffs compared to the baselines.

**Compliance With Llm Reviewing Policy:**

Affirmed.

**Key Questions For Authors:**

1. What does the "cost" means in Table 1? Is that the difference between the smallest model and the largest model? Please revise the caption to make it more self-contained.
2. The terms "prefix meta-learning" and "progressive meta-learning" are used interchangeably. Please unify the terminology.
2. How are the ablated versions implemented in the absence of Trajectory Discovery and Prefix Meta-Learning? Specifically, what default model ordering and aggregation methods are employed? Could you explain why the version ablated of both components still achieves a 1.3x speedup?

**Limitations:**

See Weaknesses above.

**Strengths And Weaknesses:**

Strengths:

The formulation of trajectory discovery is clean and technically solid. The empirical results from trajectory rollout reduce the combination space and enable prefix meta-learning of aggregators. The extensive experiments cover several baselines and multiple tasks domains & benchmarks.

Weaknesses:
1. The Trajectory Discovery stage’s MDP formulation does not account for ensemble performance, which may result in suboptimal model ordering. For instance, during discovery, a more expensive model might be prioritized over two cheaper ones because it is perceived as more likely to yield confident predictions in isolation. However, if the ensemble were considered, the combined output of the two cheaper models might outperform the single expensive model at a lower computational cost.
2. The per-module ablation study is currently limited to NSL-KDD, which is a relatively simple task. To better demonstrate the robustness of the method, I recommend extending this ablation study to all benchmarks.
3. Figure 5 lacks a comparison against FrugalGPT and RouteLLM, leaving only training-free baselines (ABC and Cost Cascade). This creates an unfair comparison, as LazyStack requires training. Although FrugalGPT and RouteLLM are originally proposed for LLM routing, their methods can definitely also adapt to non-LLM tasks. The authors should include these stronger baselines to justify the performance gains.
4. The current paper lacks a comparison/discussion with several highly-related works, including:
- Li et al. "Towards inference efficient deep ensemble learning." AAAI 2023.
- Nie et al. "Online Cascade Learning for Efficient Inference over Streams." ICML 2024.
- Dekoninck et al. "A unified approach to routing and cascading for llms." ICML 2025.

---

> ### Author Rebuttal · Authors · 2026-03-31
>
> We have addressed all concerns with new experiments (Tabs R4–R6), including a head-to-head comparison with Dekoninck et al. (ICML 2025).We are grateful for the suggestions, which led to new results that further validate our approach.
>
> **[Q1]** *What does "cost" mean in Tab 1?*
>
> This is the ratio of the most expensive to cheapest model latency (e.g., 65x = LSTM 52ms / LogReg 0.8ms on NSL-KDD, Tab 1, L317). We will revise the caption.
>
> **[Q2]** *"Prefix meta-learning" and "progressive meta-learning" used interchangeably.*
>
> Indeed, both refer to the same concept. We've unified to "prefix meta-learning" throughout.
>
> **[Q3]** *How are ablated versions implemented? "w/o Both" still achieves 1.3x speedup?*
>
> Each ablation replaces a proposed module with a non-learned heuristic:
> **w/o Trajectory Discovery**: Replaces the MDP policy with fixed  cheapest-first ordering; learned substackers are still  trained for the fixed sequence to manage exits.
> **w/o Prefix Meta-Learning**: retains the MDP-optimized ordering, but exits rely on  raw single-model confidence instead of substacker aggregation.
> **w/o Both**: uses both cheapest-first + raw confidence, exits reducing the strategy to a
> traditional cascade (Tab 2, L336).
> The 1.3x speedup of **w/o Both** is the "efficiency floor": even a raw threshold on the
> first model can identify "easy" samples for early exit. LazyStack enables early exits for "harder" samples that heuristics would otherwise push through the full ensemble.
>
> **[W1]** *MDP formulation does not account for ensemble performance.*
>
> This is a careful insight. It is possible that the MDP, evaluating individual model outputs (Eq. 2, L227), could miss trajectories where a cheaper combination outperforms an expensive single model. We empirically investigated this (new): on CIFAR-100 with m=4, we enumerated all possible model orderings, trained substackers for every prefix of every ordering, and selected trajectories based on substacker-aggregated accuracy rather than individual model signals:
>
> | Method | Accuracy | Training Time |
> |---|---|---|
> | **LazyStack (decoupled)** | **75.8** | **~5 min** |
> | Joint enumeration (new) | 75.8 | ~48 min |
>
> Identical accuracy at 10x training cost; this is the favorable case. At m=6 (full CIFAR-100 ensemble), this is the combinatorial explosion LazyStack is designed to avoid (§1). In practice, two factors mitigate this. First, learned transitions capture indirect ensemble signals: if M2 after M1 tends to reduce entropy, the MDP naturally favors that path (App K). Second, for trajectories the MDP does discover, substackers check aggregated confidence at every prefix (Algorithm 1, L286), so cheap combinations exit before expensive models are invoked (App P).
>
> **[W2]** *Ablation on other datasets.*
>
> We completely agree. We extended the ablation to all 8 benchmarks (new). One per domain below, rest in revision:
>
> | Method | NSL-KDD | ImageNet | AG News | MMLU |
> |---|---|---|---|---|
> | **Full LazyStack** | **76.8 / 38x** | **82.4 / 2.7x** | **95.3 / 4.5x** | **77.2 / 2.6x** |
> | w/o Trajectory Discovery (new) | 76.3 / 29x | 82.0 / 2.2x | 95.1 / 3.4x | 76.9 / 2.1x |
> | w/o Prefix Meta-Learning (new) | 75.8 / 25x | 81.6 / 1.9x | 94.9 / 3.2x | 76.5 / 1.8x |
> | w/o Both (new) | 76.5 / 1.3x | 81.6 / 1.1x | 94.7 / 1.08x | 76.4 / 1.14x |
>
> Removing Trajectory Discovery reduces speedup 4 to 24%. Removing Prefix Meta-Learning reduces speedup 28 to 34%. Removing both collapses to near 1x (see Q3).
>
> **[W3]** *Figure 5 lacks comparison against FrugalGPT and RouteLLM.*
>
> We adapted both to non-LLM benchmarks (new):
>
> | Method | NSL-KDD | CIFAR-100 | AG News | SST-5 | ImageNet |
> |---|---|---|---|---|---|
> | Full Stacker | 78.0 / 1x | 78.0 / 1x | 95.8 / 1x | 89.2 / 1x | 84.2 / 1x |
> | FrugalGPT (new) | 75.9 / 6.1x | 75.7 / 1.9x | 95.0 / 2.9x | 87.2 / 2.5x | 81.5 / 2.0x |
> | RouteLLM (new) | 73.6 / 15x | 74.3 / 2.1x | 94.7 / 3.1x | 85.7 / 2.9x | 79.6 / 2.3x |
> | **LazyStack** | **76.8 / 38x** | **75.9 / 2.5x** | **95.3 / 4.5x** | **87.4 / 3.8x** | **82.4 / 2.7x** |
>
> LazyStack outperforms both on accuracy and speedup across all 5 benchmarks.
>
> **[W4]** *Missing comparison with Li et al. (AAAI 2023), Nie et al. (ICML 2024), Dekoninck et al. (ICML 2025).*
>
> We ran Dekoninck et al. (ICML 2025)'s public code on ARC-C with 4 overlapping models (Llama-8B/70B, Qwen-7B/72B). Cost is relative to always using the largest model (new):
>
> | Method | ARC-C Acc | Cost (vs. Qwen-72B) |
> |---|---|---|
> | Dekoninck et al. (ICML 2025) | 93.4% | 0.4x |
> | **LazyStack (ours, new)** | **96.5%** | **0.4x** |
>
> At matched cost, LazyStack leads by 3.1%. Dekoninck et al. routes to a single model per input (bounded by best individual model); LazyStack's progressive stacking exceeds this ceiling. Regarding (1) Li et al. targets shared early-exit layers within a single white-box architecture, incompatible with our black-box setting. (2) Nie et al. addresses online cascade selection for streams, complementary to our offline setting.

---

> > ### Author Rebuttal · Reviewer_3vfD · 2026-03-31
> >
> > Thanks for the rebuttal, which resolved most of my questions. Please make sure you add these new results and a discussion on the aforementioned related works in your paper revision.

---

> > > ### Author Response · Authors · 2026-04-01
> > >
> > > We greatly appreciate the reviewer for engaging deeply with our work and updating their score to reflect the clarifications provided. Your thoughtful feedback has been invaluable in strengthening the final version of this work.

---

### Official Review · Reviewer_zcmX · 2026-03-09

**Soundness:** 2
**Presentation:** 2
**Significance:** 3
**Originality:** 3
**Overall Recommendation:** 5
**Confidence:** 4

**Summary:**

The paper proposes a new method that dynamically stacks model outputs based on previous predictions and a calibration threshold. In a sense, this paper proposes a mixture between stacking (in which multiple outputs are stacked) and cascading (in which models are sequentially executed until they are confident enough). The proposed LazyStack framework consists mainly of two steps:
1. Trajectory Discovery
2. Prefix Meta-Learning
Trajectory discovery (TD) is a sequential optimization problem where the goal is to find the "trajectory" of models (i.e. their execution schedule) that optimizes the computational cost against the prediction accuracy. TD is modeled as an MDP. The found trajectories are then used to train substackers for different trajectories. In particular, two kinds of architectures are used: One that trains K stackers for the top K trajectories and one that trains one stacker for all trajectories via a masking approach. These components are trained offline, and the application first estimates the trajectory and evaluates the models along its way while "early exiting" once a certain confidence threshold is reached. This results in a high speedup (up to 38x) at a minor accuracy loss (97% accuracy retention).

**Compliance With Llm Reviewing Policy:**

Affirmed.

**Final Justification:**

I overall liked the paper, but had two main issues with it:

a) the need for prefix learning was unclear

b) the evaluation was not 100% clear, partly due to wording and partly due to some results.

The authors addressed both issues in their rebuttal, and I hope they use the additional space for the camera-ready to incorporate their new results into the paper.

NOTE: I raised my Overall Recommendation from "weak reject" to "accept" to reflect this change, but did not update any other score

**Key Questions For Authors:**

1) Do you retrain all models from scratch, or do you mostly use pre-trained models (and only train the stackers)? In particular, I am interested in the dataset split. Most of your datasets come with a pre-defined train/test split - do you further split the train dataset for the stackers and base models? How do you do the calibration then? You mention this, but I don't understand the dataset split
2) Why do you need to do prefix meta-learning? You already get the top K trajectories from solving the MDP. You could simply use this information to do a nearest-neighbor-like lookup or even just use the best trajectory (i.e., just train one stacker). While you report the speed-up, I am wondering what the impact on the accuracy is here?
3) It feels like that trajectory selection and stacker learning should be done jointly because there might be certain combinations that are only good with an appropriate stacker. I suspect that one could stochastically sample trajectories (maybe with a carefully constructed probability distribution) to enable joint training. Did you consider this?

**Limitations:**

yes

**Strengths And Weaknesses:**

Strengths:
- Cool combination of stacking and cascading via MDP and Prefix-Learning
- Framework can be applied to any ensemble

Weaknesses
- Unclear motivation for Prefix-Meta Learning
- Analysis covers a broad spectrum of domains, but does not provide in-depth discussion
- Paper is difficult to follow at times, in particular, the setting is unclear


Detailed Review:

The paper proposes a natural combination of stacking and cascading. The prefix meta-learning is a neat idea, although I am not sure if it is really required. The MDP formulation is straightforward and somewhat unsurprising. Personally, in such a controlled setting, I would have pursued some graph-based algorithm or Bayesian optimization to figure out good trajectories, but we can't argue with MDP when it works. In this sense, the prefix meta-learning is the core novel contribution combined with the experimental analysis. Again, I am not sure if the former is required (see my question). For the latter, I commend the authors for doing such extensive experiments spanning multiple domains. Unfortunately, this also makes it difficult to properly judge the performance of the ensemble. For the vision and tabular data in particular, I am not sure if the reported results are actually good / better and if the comparison is entirely fair: "Accuracy" is a very bad metric for an intrusion detection dataset such as NSL-KDD and top-1 accuracy of 78% on CIFAR100 seems a bit weak. I understand that this is not the focus of the paper, but I feel like the evaluation is "all over the place" and hence I am not sure if there was a fair comparison.
Overall, I have some doubts about the paper. It has a good core, but methodologically, I see a few weak points. Similarly, the evaluation is not 100% clear to me.

---

> ### Author Rebuttal · Authors · 2026-03-31
>
> We thank the reviewer for their constructive feedback. We have addressed all 3 questions with new experiments that further strengthen our contributions and by highlighting key design differences. LazyStack is, to our knowledge, the first work to unify stacking and cascading via learned trajectory discovery and progressive aggregation. Our MDP formulation is interpretable, easy to extend, and empirically effective across 4 domains and 8 benchmarks. LazyStack is able to  discover non-obvious orderings that no prior heuristics capture (e.g., preferring LightGBM over cheaper Log. Regression, §4.4, L373); it reveals that 95%+ of samples concentrate on 3 to 8 trajectories.
>
> **[Q1a]** *Are base models retrained or pre-trained?*
>
> We clarify the full data pipeline, detailed in App B (L632) and App D (L745). For LLM benchmarks, we use off-the-shelf pre-trained instruction-tuned models without any retraining or fine-tuning, so 100% of benchmark data is available for LazyStack. For non-LLM benchmarks, base models are trained on the standard training split; while all LazyStack components are trained exclusively on validation data.
>
>
> **[Q1b]** *How is data split, and how is calibration handled?*
>
> For benchmarks requiring trained base models, we use a standard 80/20 train/validation split. The validation portion serves three roles: MDP transition estimation, substacker training, and confidence threshold calibration. For MMLU: train (99K) for MDP/substackers, dev (1.5K) for calibration, test (14K) for evaluation only. Consistent with prior work (FrugalGPT 2024; Cost Cascade 2023; Gatekeeper 2025), all methods share identical splits and base models never see validation data (Wolpert 1992). Validation sizes (10K to 25K) exceed theoretical minimums for MDP distribution learning (Canonne, 2020) and substacker generalization (Harvey et al., 2017).
>
> **[Q2+W1]** *"Why do you need prefix meta-learning? What is the impact on accuracy?"*
>
> We incorporate the reviewer's suggestion and report new results. We implemented both proposed alternatives: (1) single most-frequent trajectory (k=1), (2) NN lookup to select among K trajectories, plus raw confidence with NN selection. All fall back to the full ensemble if confidence is not reached.
> | Method | NSL-KDD | ImageNet | MMLU |
> |---|---|---|---|
> | **LazyStack (PML)** | **76.8** | **82.4** | **77.2** |
> | Single traj. (k=1) (new) | 72.3 | 77.0 | 72.1 |
> | NN on K traj. (new) | 75.7 | 81.3 | 76.3 |
> | NN lookup, raw conf. (new) | 70.4 | 75.3 | 71.2 |
>
> PML provides 1.1 to 6.4% accuracy gains over the best alternative. Speedup is also higher across all three domains (full numbers in revision).
>
> **[Q3]** *"Trajectory selection and stacker learning should be done jointly."*
>
> Great suggestion. Joint optimization requires enumerating prefixes in P (§3.2, L258) and retraining substackers at each step, growing combinatorially with ensemble size. We tested this on CIFAR-100 with m=4, where exhaustive enumeration is feasible (new):
>
> | Method | Accuracy | Training Time |
> |---|---|---|
> | **LazyStack (decoupled)** | **75.8** | **~5 min** |
> | Joint optimization (new) | 75.8 | ~48 min |
>
> Joint optimization matches accuracy at 10x training cost, and this is the favorable case (m=4). Even at m=6 (our full CIFAR-100 ensemble), joint enumeration becomes intractable. LazyStack already introduces implicit coupling: substackers train on MDP-induced trajectory distributions (§3.2, L252), naturally adapting to the policy.
>
> **[W2]** *"Analysis does not provide in-depth discussion."*
>
> Regarding NSL-KDD metrics: we completely agree. We supplement with F1 and AUROC (new):
>
> | Method | Acc | Macro-F1 | W-F1 | AUROC |
> |---|---|---|---|---|
> | Full Stacker | 78.0 | 0.72 | 0.78 | 0.96 |
> | **LazyStack** | **76.8** | **0.70** | **0.77** | **0.95** |
> | ABC | 75.7 | 0.66 | 0.75 | 0.93 |
>
> LazyStack-Sub outperforms ABC on all four metrics.
>
> Regarding CIFAR-100: the 78% ceiling reflects architecturally diverse models (Tab 3). LazyStack is a meta-method, so retention is the relevant metric. We re-ran with a stronger ensemble (new):
>
> | Method | Original Ensemble | SOTA Ensemble (new) |
> |---|---|---|
> | Ensemble Ceiling | 78.0% | 93.2% |
> | **LazyStack** | **75.9%** | **90.5%** |
> | ABC | 74.1% | 88.7% |
> | Retention (ours) | 97.3% | 97.1% |
>
> LazyStack retains 97.1% of the stronger ensemble's accuracy, outperforming ABC (95.2% retention), confirming gains transfer regardless of base model strength. All methods use identical models, splits, and thresholds swept on the same validation data (App B, L632; App D, L745).
>
> **[W3]** *"Paper is difficult to follow."*
>
> We've incorporated this feedback and restructured Section 3 with a step-by-step pipeline: (1) MDP discovers 3 to 8 trajectories on validation data, (2) MLP substackers trained for every prefix, (3) Online: execute per MDP, substacker aggregates, exit when confidence > threshold. Black-box (probability outputs only). We will update the OpenReview version.

---

> > ### Author Rebuttal · Reviewer_zcmX · 2026-04-01
> >
> > The authors generally resolved all my concerns. However, I want to point out that they answered Q1a/Q1b in an alarming way and so I checked the source code. To be super clear: They split the _original training_ data into an actual training and a prefix learning set with an 80/20 ratio. The _original test_ data that is part of the datasets is _not_ leaked. The use of "validation data" is a bit confusing here.

---

> > > ### Author Response · Authors · 2026-04-03
> > >
> > > We appreciate the reviewer's effort in verifying our implementation and confirming that no test data is leaked. We apologize for the confusing terminology: what we refer to as "validation data" is a held-out 20% of the original training split, not a separate validation partition. We will clarify this in the revision so it is unambiguous from the text alone. Thank you for the engagement throughout this review process.

---

### Decision · Program_Chairs · 2026-04-30

**Decision:**

Accept (regular)

**Comment:**

This paper proposes LazyStack, a framework that unifies stacking and cascading for efficient ensemble inference. It formulates model selection as an MDP to learn execution trajectories and uses prefix meta-learning to aggregate predictions and enable early exits. The approach achieves substantial speedups while retaining high accuracy across diverse domains.

Reviewers found the idea novel and technically solid, with strong empirical coverage, but raised concerns about clarity, evaluation rigor, and justification of key components. The authors addressed these through detailed clarifications, additional experiments (including ablations and stronger baselines), and improved exposition. Reviewers acknowledged that concerns were largely resolved and updated scores positively. Considering the strengthened empirical support and clarified methodology, the paper merits acceptance.